# Dystroglycan is a scaffold for extracellular axon guidance decisions

**L Bailey Lindenmaier[1], Nicolas Parmentier[2], Caiying Guo[3], Fadel Tissir[2], Kevin M Wright[1]***

[1]Vollum Institute, Oregon Health & Science University, Portland, United States; [2]Institiute of Neuroscience, Université Catholique de Louvain, Brussels, Belgium; [3]Janelia Research Campus, Howard Hughes Medical Institute, Ashburn, United States

**Abstract** Axon guidance requires interactions between extracellular signaling molecules and transmembrane receptors, but how appropriate context-dependent decisions are coordinated outside the cell remains unclear. Here we show that the transmembrane glycoprotein Dystroglycan interacts with a changing set of environmental cues that regulate the trajectories of extending axons throughout the mammalian brain and spinal cord. Dystroglycan operates primarily as an extracellular scaffold during axon guidance, as it functions non-cell autonomously and does not require signaling through its intracellular domain. We identify the transmembrane receptor Celsr3/Adgrc3 as a binding partner for Dystroglycan, and show that this interaction is critical for specific axon guidance events in vivo. These findings establish Dystroglycan as a multifunctional scaffold that coordinates extracellular matrix proteins, secreted cues, and transmembrane receptors to regulate axon guidance.

DOI: https://doi.org/10.7554/eLife.42143.001

## Introduction

During neural circuit development, extending axons encounter distinct combinations of cues and growth substrates that guide their trajectory. These cues can be attractive or repulsive, secreted and/or anchored to cell membranes, and signal through cell surface receptors on the growth cones of axons (*Kolodkin and Tessier-Lavigne, 2011*). Receptors also recognize permissive and non-permissive growth substrates formed by the extracellular matrix (ECM), surrounding cells, and other axons (*Raper and Mason, 2010*). While many cues and receptors that direct axon guidance have been identified, our understanding of how cues in the extracellular space are organized and interpreted by growing axons is far from complete.

In a previous forward genetic screen for novel mediators of axon guidance, we identified two genes, *Isoprenoid Synthase Domain Containing* (*Ispd*) and *Beta-1,4-glucuronyltransferase 1* (*B4gat1*, formerly known as *B3gnt1*), that are required for the functional glycosylation of the transmembrane protein Dystroglycan (*Wright et al., 2012*). Dystroglycan is comprised of a heavily glycosylated extracellular α-subunit that is non-covalently linked to its transmembrane β-subunit (*Barresi and Campbell, 2006*). The mature 'matriglycan' epitope on α-Dystroglycan is required for its ability to bind extracellular proteins that contain Laminin G (LG) domains, including Laminins, Perlecan, Agrin, Pikachurin, Neurexin, and Slit (*Campanelli et al., 1994*; *Gee et al., 1994*; *Ibraghimov-Beskrovnaya et al., 1992*; *Peng et al., 1998*; *Sato et al., 2008*; *Sugita et al., 2001*; *Wright et al., 2012*; *Yoshida-Moriguchi and Campbell, 2015*; *Yoshida-Moriguchi et al., 2010*). The intracellular domain of β-Dystroglycan interacts with the actin binding proteins Dystrophin and Utrophin, and can also function as a scaffold for ERK/MAPK and Cdc42 pathway activation (*Batchelor et al., 2007*; *Ervasti and Campbell, 1993*; *James et al., 1996*; *Spence et al., 2004*). Therefore, Dystroglycan

*For correspondence:
wrighke@ohsu.edu

serves as a direct link between the ECM and pathways involved in cytoskeletal remodeling and filo-podial formation, suggesting that it can function as an adhesion receptor to regulate cell motility and migration (*Moore and Winder, 2010*). However, this has not been examined in vivo.

Mutations that result in hypoglycosylation of α-Dystroglycan result in a loss of ligand binding capacity and cause a form of congenital muscular dystrophy (CMD) referred to as dystroglycanop-athy. Severe forms of this disorder are accompanied by neurodevelopmental abnormalities including type II lissencephaly, hydrocephalus, brainstem and hindbrain hypoplasia, ocular dysplasia, and white matter defects (*Godfrey et al., 2011*). We previously found that glycosylated Dystroglycan regulates axon guidance in the developing spinal cord and in the optic chiasm by maintaining the basement membrane as a permissive growth substrate and organizing the extracellular localization of Slit pro-teins in the floor plate (*Clements and Wright, 2018*; *Wright et al., 2012*). However, there are a number of outstanding questions about the role of Dystroglycan in axon guidance: Does Dystrogly-can function in axons as an adhesion receptor? Is Dystroglycan required for the formation of other axon tracts in the mammalian nervous system? Does Dystroglycan bind additional LG-domain con-taining proteins important for axon guidance?

Here, we provide genetic evidence that Dystroglycan operates non-cell autonomously and relies on its extracellular scaffolding function to regulate the development of multiple axon tracts in the spinal cord and brain. We identify a novel interaction between Dystroglycan and Celsr3 (Adgrc3), an LG-domain containing transmembrane receptor that regulates axon guidance in the brain, spinal cord, and peripheral nervous system (*Chai et al., 2014*; *Onishi et al., 2013*; *Tissir et al., 2005*; *Zhou et al., 2008*). Using genome editing to generate a *Celsr3* mutant that is unable to bind Dystro-glycan (*Celsr3$^{R1548Q}$*), we show that this interaction is specifically required to direct the anterior turn-ing of post-crossing spinal commissural axons in vivo. These results define a novel interaction between Dystroglycan and Celsr3, and establish Dystroglycan as a multifunctional regulator of axon guidance throughout the nervous system via its coordination of multiple ECM proteins, secreted cues, and transmembrane receptors.

## Results

### Dystroglycan functions non-cell autonomously as an extracellular scaffold to guide commissural axons

We have previously shown that defective glycosylation of Dystroglycan or conditional deletion of *Dystroglycan* throughout the epiblast results in defective axon tract formation in the developing spi-nal cord and visual system. We found that Dystroglycan is required to maintain the basement mem-brane as a permissive growth substrate and for the proper extracellular localization of the secreted axon guidance cue Slit (*Clements and Wright, 2018*; *Wright et al., 2012*). However, we have not tested whether Dystroglycan has a cell-autonomous role in regulating the guidance of spinal com-missural axons. Examination of E12.5 spinal cord sections shows that in addition to its enrichment in the floor plate and the basement membrane (*Figure 1A* inset, arrows), Dystroglycan protein was detected in spinal commissural axons (*Figure 1A*, *Figure 1—figure supplement 1A*). The specificity of the Dystroglycan expression pattern was confirmed by showing its loss in mice in which the intra-cellular domain of Dystroglycan is genetically deleted (*Figure 1—figure supplement 1B*). In cultured e12.5 commissural axons, Dystroglycan was expressed throughout the axon, including the growth cone (arrows, *Figure 1B*). These results show that Dystroglycan is expressed in both commissural axons and the surrounding environment through which they navigate.

Based on its association with the actin-binding proteins Dystrophin and Utrophin and its ability to regulate filopodial formation *via* ERK/MAPK and Cdc42 activation, we hypothesized that Dystrogly-can could function within commissural axons as an adhesion receptor *in vivo*. To test this, we per-formed DiI labeling in open-book e12.5 spinal cord preparations. In control open book preparations, an average of $97.62 \pm 2.38\%$ of injection sites showed normal floor plate crossing and anterior turn-ing (*Figure 1C,G*). In agreement with our previous findings, normal floorplate crossing and turning was observed in only $3.03\pm3.03\%$ of injection sites from mice lacking Dystroglycan throughout the developing spinal cord (*Dag1$^{F/-}$;Sox2$^{Cre}$*) (*Figure 1D,G*). This phenotype is fully penetrant, and all of the abnormal injections sites in *Dag1$^{F/-}$;Sox2$^{Cre}$* mutants exhibited both stalling within the floorplate and an anterior-posterior (AP) randomization of post-crossing axonal trajectory. We next examined

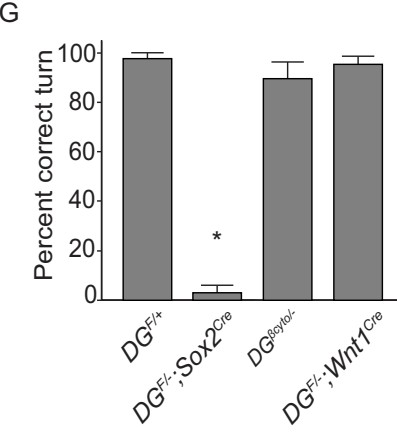

**Figure 1.** Dystroglycan functions non-cell autonomously to guide spinal commissural axons. (**A**) Immunostaining of E12.5 spinal cord shows Dystroglycan protein (magenta, left panel) expression in commissural axons (L1, green, middle panel). In the high magnification insets, arrows indicate the enriched expression of Dystroglycan in the basement membrane of the spinal cord proximal to the axons. (**B**) Commissural neurons from E12 dorsal spinal cord cultured for two days in vitro (2DIV) were stained with antibodies to Dystroglycan (magenta, left panel), TUJ1 (green, middle panel).
*Figure 1 continued on next page*

| Genotype | DG status | # embryos analyzed | # injection sites | % normal (SEM) | phenotype |
|---|---|---|---|---|---|
| *Dag1^Flox/+* | present | 6 | 49 | 97.62 (2.38) | normal |
| *Dag1^Flox/-;Sox2^Cre* | deleted from epiblast | 3 | 18 | 3.03 (3.03) | stalling; AP randomization |
| *Dag1^βcyto/-* | lacking intracellular domain | 3 | 34 | 89.52 (6.75) | normal |
| *Dag1^Flox/-; Wnt1^Cre* | deleted from comm axons | 8 | 59 | 95.31 (3.29) | normal |

*Figure 1 continued*

Dystroglycan is present throughout the cell body, axon and growth cone (arrow). (**C–F**) DiI injections in open-book preparations of E12 spinal cords were used to examine the trajectory of commissural axons. In controls (**C**), axons extend through the floor plate, then execute an anterior turn (n=6 animals, 49 total injection sites). In *Dag1$^{F/-}$;Sox2$^{Cre}$* mice (**D**), axons stall within the floor plate and post-crossing axons exhibit anterior-posterior randomization (n=3 animals, 18 total injection sites). (**E**) Commissural axons in mice lacking the intracellular domain of Dystroglycan (*Dag1$^{βcyto/-}$*) show normal crossing and anterior turning (n=3 animals, 34 total injection sites). Conditional deletion of *Dystroglycan* from commissural neurons in *Dag1$^{F/-}$; Wnt1$^{Cre}$* mice (**F**) did not affect floor plate crossing or anterior turning (n=8 animals, 59 total injection sites). Higher magnification insets for each image show the anterior (top) and posterior (bottom) trajectories of post-crossing commissural axons. (**G**) Quantification of open book preparations. On average, 97.62 ± 3.39% of controls, 3.03 ± 4.80% of *Dag1$^{F/-}$;Sox2$^{Cre}$* mutants, 89.52 ± 4.80% of *Dag1$^{βcyto/-}$* mutants, and 95.31 ± 2.94% of *Dag1$^{F/-}$; Wnt1$^{Cre}$* mutants showed normal crossing and anterior turning. All of the *Dag1$^{F/-}$;Sox2$^{Cre}$* mutants with turning defects also showed stalling within the floor plate. *p< 0.001, one-way ANOVA, Tukey's *post hoc* test. Scale bar = 100µm (**A**), 10µm (**B**) and 50µm (**F–H**).

DOI: https://doi.org/10.7554/eLife.42143.002

The following figure supplement is available for figure 1:

**Figure supplement 1.** Analysis of Dystroglycan expression and commissural axon phenotypes in spinal cord sections.

DOI: https://doi.org/10.7554/eLife.42143.003

commissural axons in which the intracellular domain of Dystroglycan is deleted (*Dag1$^{βcyto/-}$*), rendering it unable to bind dystrophin/utrophin or initiate ERK/MAPK or Cdc42 signaling (*Satz et al., 2009*). To our surprise, 89.52±6.75% of injection sites in *Dag1$^{βcyto/-}$* mutants showed both normal floorplate crossing and anterior turning (*Figure 1E,G*), suggesting that the intracellular domain of Dystroglycan is dispensable for commissural axon guidance. To further test for a cell-autonomous role for Dystroglycan during axon guidance, we examined mice in which *Dystroglycan* is conditionally deleted from commissural axons (*Dag1$^{F/-}$;Wnt1$^{Cre}$*). 95.31±3.29% of injection sites in *Dag1$^{F/-}$; Wnt1$^{Cre}$* open book preparations displayed normal commissural axon growth and post-crossing anterior turning (*Figure 1F,G*).

We also examined spinal commissural axons by staining E12.5 spinal cord sections with antibodies to L1 or Robo1 and Robo2. As we have previously shown, post-crossing commissural axons in *Dag1$^{F/-}$;Sox2$^{Cre}$* mutants exhibit abnormal bundling and disruptions along the ventrolateral funiculus (*Figure 1—figure supplement 1D*). In contrast, he ventrolateral funiculus appears normal in both *Dag1$^{βcyto/-}$* and *Dag1$^{F/-}$;Wnt1$^{Cre}$* mutants (*Figure 1—figure supplement 1E–F*), confirming the results obtained with open book preparations. Taken together, these results support a model that Dystroglycan functions non-cell autonomously as an extracellular scaffold to guide commissural axons in vivo.

## Dystroglycan is required for axon tract development in the forebrain

We next sought to determine whether loss of functional Dystroglycan also affected the formation of axon tracts in other regions of the developing nervous system. At E14.5, Dystroglycan is expressed by neuroepithelial cells and is enriched in the basement membrane surrounding the brain. It is also present in the ventral telencephalon, particularly in axons in the thalamus and the developing internal capsule, which is comprised of ascending thalamocortical and descending corticothalamic axons (*Figure 2—figure supplement 1A–B*). Therefore, we hypothesized that Dystroglycan may be required for axon tract development in the forebrain. In *Ispd$^{L79*/L79*}$* mutants, which lack glycosylated Dystroglycan, and *Dag1$^{F/-}$;Sox2$^{Cre}$* mutants, in which *Dystroglycan* is deleted throughout the epiblast, we observed severe defects in multiple forebrain axon tracts (*Figure 2*, *Figure 2—figure supplement 1D,F*). Abnormalities included fasciculated axons in the upper layers of the cortex (arrows), a large, swirling bundle of axons in the ventral telencephalon (asterisk), and a large axonal projection inappropriately exiting through the ventral diencephalon (arrowheads) (*Figure 2A–C*). These defects were somewhat variable in their severity, but were fully penetrant and all three of these defects were observed in all mutants.

To better understand the nature of the axonal defects in *Ispd$^{L79*/L79*}$* and *Dag1$^{F/-}$;Sox2$^{Cre}$* mutants, we used anterograde tract tracing. DiI labeling of thalamocortical axons (TCAs) in controls showed that axons cross the diencephalon-telencephalon boundary (DTB), extend dorsolaterally through the ventral telencephalon, and cross the pallial-subpallial boundary (PSPB) before turning medially to extend along the intermediate zone of the cortex (*Figure 2D,J*). In contrast, TCAs in both *Ispd$^{L79*/L79*}$* and *Dag1$^{F/-}$;Sox2$^{Cre}$* mutants largely failed to cross the DTB, and instead extended

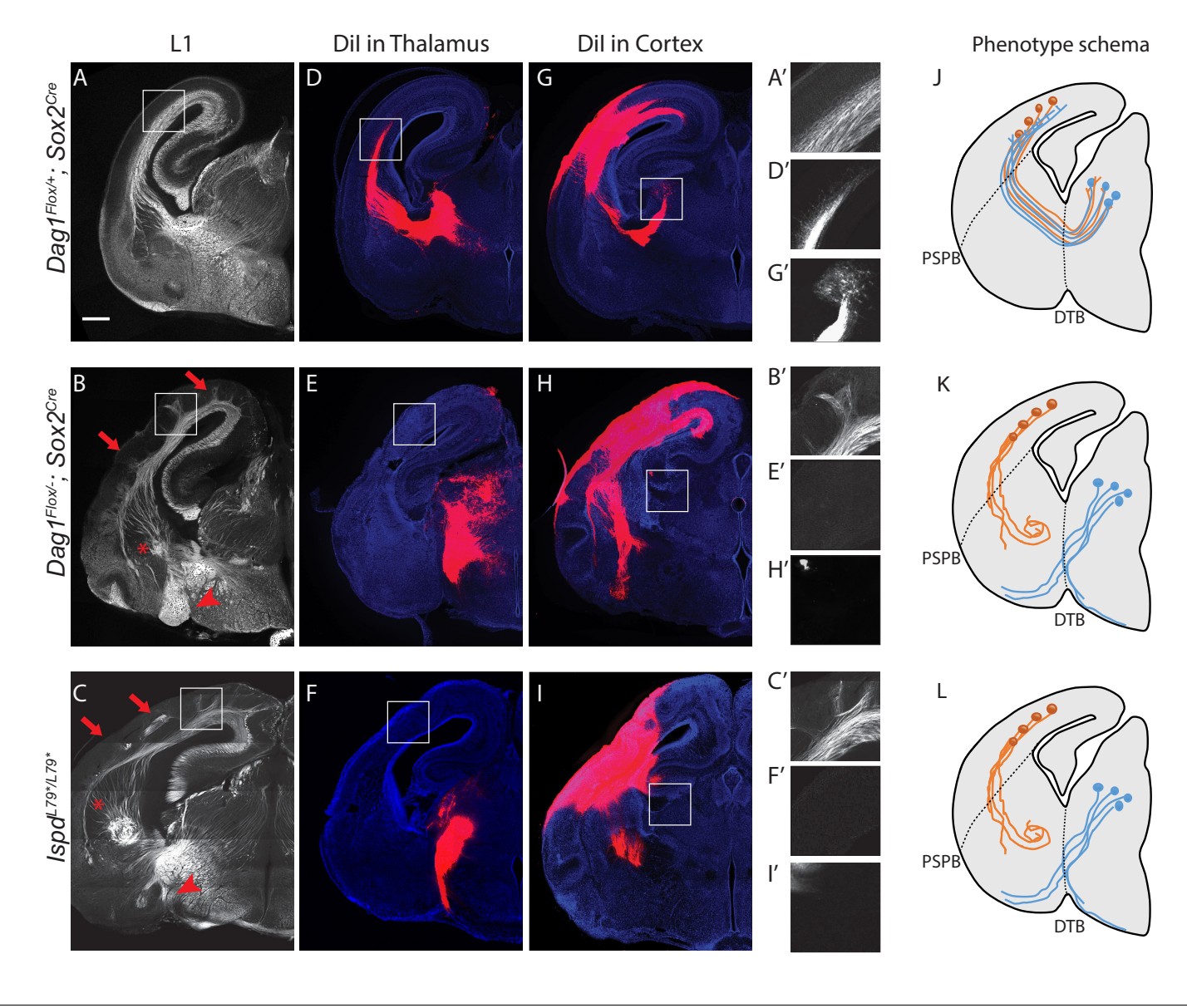

**Figure 2.** Dystroglycan is required for axon tract formation in the forebrain. (**A**) L1 immunohistochemistry on P0 brain sections from $Dag1^{F/+};Sox2^{Cre}$ controls (n = 3 animals) labels descending CTAs and ascending TCAs in the internal capsule. In $Dag1^{F/-};Sox2^{Cre}$ (n = 4 animals) (**B**) and $Ispd^{L79*/L79*}$ (n = 5 animals) (**C**) mutants, the internal capsule is highly disorganized, with axons projecting into the upper layers of the cortex (red arrows), forming ectopic bundles in the ventral telencephalon (red asterisks), and abnormal projections extending ventrally (red arrowheads). High magnification insets show L1 +axons in the intermediate zone of the cortex of controls (**A'**) and ectopic axonal projections into the upper cortical layers in $Dag1^{F/-};Sox2^{Cre}$ (**B'**) and $Ispd^{L79*/L79*}$ (**C'**) mutants. DiI injection in the thalamus of $Dag1^{F/+};Sox2^{Cre}$ controls (n = 4 animals) labels TCAs as they cross the DTB, extend through the ventral telencephalon, across the PSPB, and into the intermediate zone of the cortex. In $Dag1^{F/-};Sox2^{Cre}$ (n = 4 animals) (**E**) and $Ispd^{L79*/L79*}$ (n = 4 animals) (**F**) mutants, TCAs fail to cross the DTB, and instead project ventrally out of the diencephalon. High magnification insets show DiI-labeled TCAs extending into the intermediate zone of the cortex of controls (**D'**), and a lack of labeled TCAs in the cortex of $Dag1^{F/-};Sox2^{Cre}$ (**E'**) and $Ispd^{L79*/L79*}$ (**F'**) mutants. DiI injection in the cortex of $Dag1^{F/+};Sox2^{Cre}$ controls (n = 3 animals) labels CTAs as they extend across the PSPB, through the ventral telencephalon, and across the DTB into the thalamus. CTAs in $Dag1^{F/-};Sox2^{Cre}$ (n = 4 animals) (**H**) and $Ispd^{L79*/L79*}$ (n = 5 animals) (**I**) mutants fail to cross the PSPB or take abnormal trajectories through the ventral telencephalon. High magnification insets show DiI-labeled CTAs extending into the thalamus in controls (**G'**), and a lack of labeled CTAs in the thalamus of $Dag1^{F/-};Sox2^{Cre}$ (**I'**) and $Ispd^{L79*/L79*}$ (**F'**) mutants. (**J–L**) Schematic summarizing CTA (brown) and TCA (blue) axon trajectories in controls (**J**), $Dag1^{F/-};Sox2^{Cre}$ (**K**) and $Ispd^{L79*/L79*}$ (**L**). Scale bar = 500 µm.

DOI: https://doi.org/10.7554/eLife.42143.004

The following figure supplement is available for figure 2:

**Figure supplement 1.** Anterior commissure, lateral olfactory tract and corpus callosum phenotypes in $Ispd^{L79*/L79*}$ mutants.

*Figure 2 continued on next page*

*Figure 2 continued*

DOI: https://doi.org/10.7554/eLife.42143.005

ventrally out of the diencephalon, often joining the optic tract (*Figure 2E,F,K,L*). Occasionally, TCAs take a more rostral route through the ventral telecephalon in an abnormal trajectory, where they eventually turn and enter the cortex. These aberrant TCAs then extend into the upper layers of the cortex in large fascicles rather than remaining in the intermediate zone (data not shown).

DiI injections in the cortex of controls labeled corticothalamic axons (CTAs) that project across the PSPB, then execute a ventromedial turn to project through the ventral telencephalon before turning dorsomedially across the DTB into the thalamus (*Figure 2G,J*). DiI labeling in $Ispd^{L79*/L79*}$ and $Dag1^{F/-};Sox2^{Cre}$ mutants indicated that many CTAs fail to cross the PSPB. Axons that do cross the PSPB stall or take abnormal trajectories through the ventral telencephalon (*Figure 2H,I,K,L*). Few CTAs in $Ispd^{L79*/L79*}$ and $Dag1^{F/-};Sox2^{Cre}$ mutants were able to correctly navigate through the internal capsule to arrive at the thalamus.

In addition to the defects in TCAs and CTAs, other axon tracts within the developing forebrain were malformed in $Ispd^{L79*/L79*}$ and $Dag1^{F/-};Sox2^{Cre}$ mutants. The anterior commissure was frequently diminished in $Ispd^{L79*/L79*}$ mutants (*Figure 2—figure supplement 1E,F*). The lateral olfactory tract (LOT), which contains axons projecting from the olfactory bulb to cortical targets, normally forms directly beneath the pial surface of the ventrolateral rostral forebrain (arrowheads, *Figure 2—figure supplement 1C,E*). In $Ispd^{L79*/L79*}$ mutants, the LOT was consistently abnormal, often projecting deeper into the ventrolateral forebrain (arrowheads, *Figure 2—figure supplement 1D,F*). We cannot exclude that there are fewer axons in the anterior commissure or the LOT in $Ispd^{L79*/L79*}$ mutants, as the extreme disorganization of these axons tracts prevents accurate quantification. In contrast, the corpus callosum in $Ispd^{L79*/L79*}$ mutants appears largely normal, despite the prominent number of axons projecting inappropriately into the upper layers of the cortex (*Figure 2—figure supplement 1D,F*). Taken together, these results show that glycosylated Dystroglycan is required for proper development of multiple axon tracts in the forebrain.

## Dystroglycan functions non-cell autonomously to guide thalamocortical and corticothalamic axons

Where does Dystroglycan function during forebrain axon tract development? As ascending TCAs and descending CTAs form the internal capsule, they interact with several intermediate targets along their trajectory (*Figure 3A,A'*). TCAs are guided ventrolaterally across the DTB by Isl1 +guidepost cells, then extend through a permissive 'corridor' in the ventral telencephalon formed by lateral ganglionic eminence (LGE) derived cells (*Feng et al., 2016*; *López-Bendito et al., 2006*; *Métin and Godement, 1996*). TCAs contact CTAs at the PSPB, then track along them within the intermediate zone, where they pause for several days before invading the cortical layers (*Blakemore and Molnár, 1990*; *Catalano and Shatz, 1998*; *Chen et al., 2012*). Descending CTAs extend in the opposite direction, first crossing the PSPB, then extending medially through the ventral telencephalon to the DTB along TCAs, where they turn dorsally into the thalamus (*De Carlos and O'Leary, 1992*; *Molnár and Cordery, 1999*).

Based on immunohistochemistry, Dystroglycan is expressed on both axons in the internal capsule, as well as intermediate targets/guidepost cells. To identify the specific cellular population in which Dystroglycan is required during internal capsule formation, we took advantage of *Dystroglycan* conditional mutants. We first examined $Dag1^{F/-};Foxg1^{Cre}$ mutants, in which Dystroglycan is deleted in neuroepithelial cells and their progeny throughout the dorsal and ventral telencephalon. This includes CTAs and guidepost cells in the ventral telencephalon, but not the developing thalamus or TCAs (*Figure 3B'*). Using immunostaining and DiI labeling, we found that both TCAs and CTAs took abnormal trajectories in $Dag1^{F/-};Foxg1^{Cre}$ mutants that were similar to those observed in $Dag1^{F/-};Sox2^{Cre}$ and $Ispd$ mutants (arrowheads, *Figure 3B*, *Figure 3—figure supplement 1B,B'*). This phenotype was milder than $Dag1^{F/-};Sox2^{Cre}$ mutants, with some TCAs reaching the thalamus, but was fully penetrant, with all $Dag1^{F/-};Foxg1^{Cre}$ mutants showing a similar phenotype. To test whether Dystroglycan functions within TCAs, we utilized $Dag1^{F/-};Gbx2^{CreERT2}$ mutants, in which tamoxifen administered at E10 results in recombination throughout the developing thalamus (*Figure 3—figure*

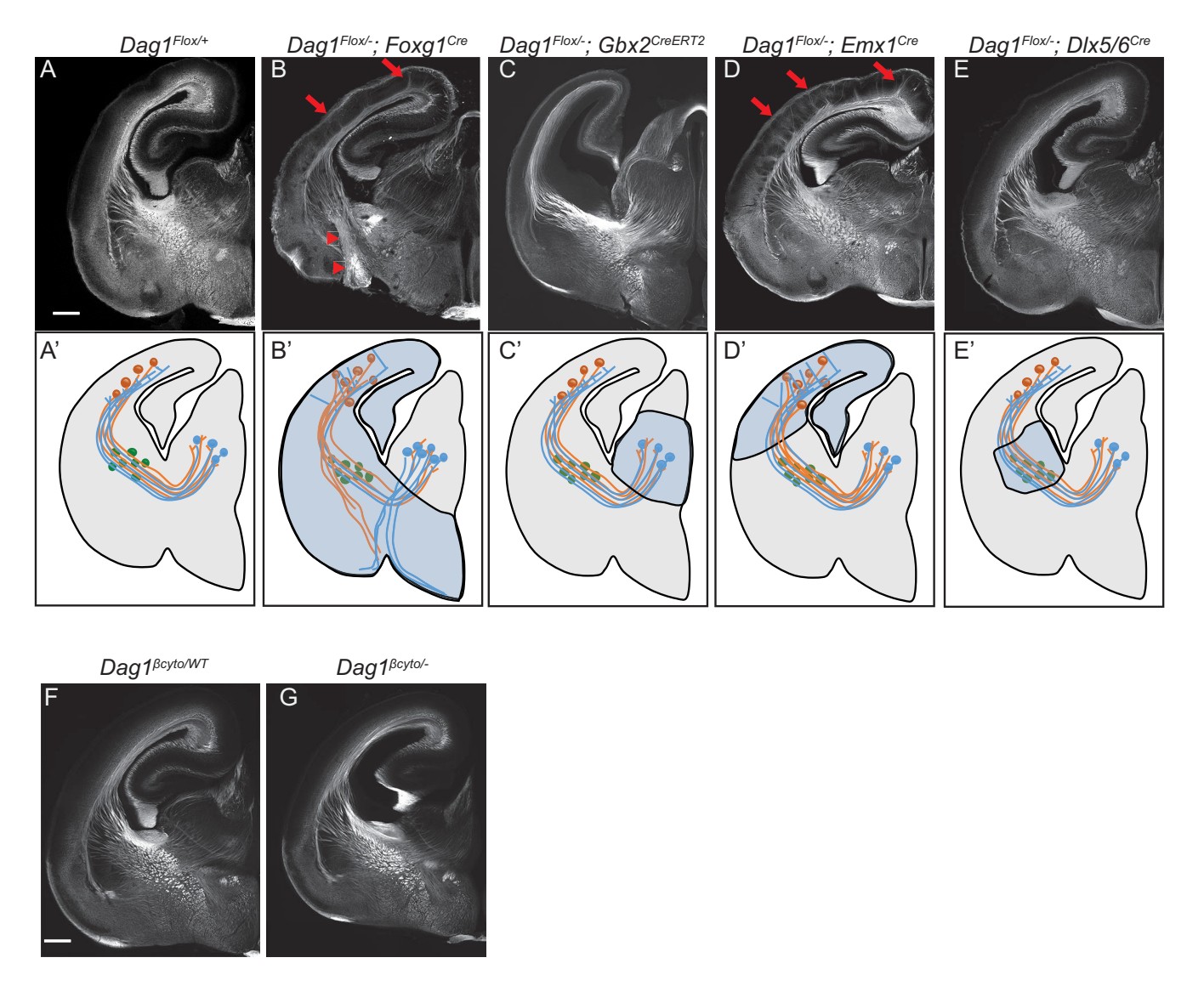

**Figure 3.** Dystroglycan is required in ventral telencephalon neuroepithelial cells to guide corticothalamic and thalamocortical axons. L1 staining of P0 brain sections from $Dag1^{F/+}$ controls (n = 3 animals) (**A, F**), $Dag1^{F/-};Foxg1^{Cre}$ (n = 4 animals) (**B**), $Dag1^{F/-};Gbx2^{CreERT2}$ (n = 4 animals) (**C**), $Dag1^{F/-};Emx1^{Cre}$ (n = 3 animals) (**D**), $Dag1^{F/-};Dlx5/6^{Cre}$ (n = 3 animals) (**E**), and $Dag1^{βcyto/-}$ (n = 5 animals) (**G**). A'-E' illustrate the recombination patterns in each Cre/CreERT2 line in the blue shaded area. Deletion of $Dystroglycan$ throughout the neuroepithelium of the dorsal and ventral telencephalon in $Dag1^{F/-};Foxg1^{Cre}$ mutants (**B, B'**) results in abnormal projections in the internal capsule (red arrowheads) and abnormal axonal projections into the upper layers of the cortex (red arrows). Deletion of $Dystroglycan$ from the neuroepithelium of the dorsal telencephalon with $Emx1^{Cre}$ mutants (**D**) results in abnormal axonal projections into the upper layers of the cortex (red arrows), but normal internal capsule formation. Deletion of $Dystroglycan$ from the thalamus with $Gbx2^{CreERT2}$ (**C**) or 'corridor' cells with $Dlx5/6^{Cre}$ (**E**) did not affect axon guidance. Deletion of the intracellular domain of Dystroglycan in $Dag1^{βcyto/-}$ mutants (**G**) did not affect formation of the internal capsule compared to control littermates (**F**). A-G Scale bar = 500 µm.
DOI: https://doi.org/10.7554/eLife.42143.006

The following figure supplements are available for figure 3:

**Figure supplement 1.** DiI labeling of CTAs and TCAs in $Dystroglycan$ conditional mutants.
DOI: https://doi.org/10.7554/eLife.42143.007

**Figure supplement 2.** Recombination pattern in $Gbx2^{CreERT2}$ and $Emx1^{Cre}$ mice.
DOI: https://doi.org/10.7554/eLife.42143.008

*supplement 2A*). Based on L1 staining and DiI labeling of TCAs and CTAs, the internal capsule is normal *Dag1$^{F/-}$;Gbx2$^{CreERT2}$* mutants (*Figure 3C*, *Figure 3—figure supplement 1C,C'*). Taken together, these results suggest that Dystroglycan is required in the telencephalon and not in TCAs during internal capsule formation.

To further dissect the role of Dystroglycan in the telencephalon, we examined *Dag1$^{F/-}$;Emx1$^{Cre}$* mutants in which recombination occurs in neuroepithelial cells and their progeny in the dorsal but not ventral telencephalon (*Figure 3D'*, *Figure 3—figure supplement 2B*). *Dag1$^{F/-}$;Emx1$^{Cre}$* mutants exhibit significant cortical lamination defects, consistent with the known role of Dystroglycan in regulating cortical migration by maintaining integrity of the neuroepithelial scaffold (data not shown) (*Moore et al., 2002*; *Myshrall et al., 2012*; *Satz et al., 2008*). Despite the abnormal cell body positioning of subcortically-projecting deep layer neurons, their ability to extend axons across the PSPB, through the internal capsule, and into the thalamus appeared unaffected (*Figure 3D*, *Figure 3—figure supplement 1D*). These results suggest that Dystroglycan is not required in CTAs during internal capsule formation. Reciprocal projections from TCAs were likewise able to extend normally through the internal capsule in *Dag1$^{F/-}$;Emx1$^{Cre}$* mutants, but upon entering the cortex, formed fasciculated bundles that projected into the upper levels of the cortex (*Figure 3D*, arrows, *Figure 3—figure supplement 1D'*). This phenotype was fully penetrant, with all *Dag1$^{F/-}$;Emx1$^{Cre}$* mutants showing a similar phenotype. We interpret this as a secondary effect of the migration defects in the cortex *Dag1$^{F/-}$; Emx1$^{Cre}$* mutants, as it resembles phenotypes seen when subplate neurons are mislocalized to the upper layers of the cortex (*Molnár et al., 1998*; *Rakić et al., 2006*). Finally, we examined the effect of deleting Dystroglycan with *Dlx5/6$^{Cre}$*, which recombines in LGE-derived postmitotic neurons in the ventral telecephalon, including Isl1 +guidepost neurons that migrate laterally to form the permissive 'corridor' for TCAs (*Figure 3E'*). CTAs and TCAs in *Dag1$^{F/-}$;Dlx5/6$^{Cre}$* mutants extended through the internal capsule and into their target regions normally (*Figure 3E*, *Figure 3—figure supplement 1E and E'*), demonstrating that Dystroglycan is not required in guidepost cells in the internal capsule.

We also tested whether forebrain axon guidance required signaling through the intracellular domain of Dystroglycan. L1 staining shows that the internal capsule, anterior commissure, lateral olfactory tract, and corpus callosum were all normal in *Dag1$^{\beta cyto/-}$* mutants (*Figure 3G*, data not shown), demonstrating that intracellular signaling by Dystroglycan is completely dispensable for forebrain axon guidance. Collectively, we conclude that Dystroglycan is not required in CTAs (*Emx1-Cre*), TCAs (*Gbx2$^{Cre}$*), or 'corridor' cells (*Dlx5/6$^{Cre}$*), but is required in neuroepithelial cells in the ventral telencephalon (*Foxg1$^{Cre}$*). Taken together with our results in spinal commissural axons, these data support a model in which Dystroglycan functions non-cell autonomously as an extracellular scaffold to guide axon tract formation in multiple CNS regions.

## Dystroglycan binds to the axon guidance receptor Celsr3

What are the relevant binding partners for glycosylated Dystroglycan during axon guidance? The majority of interacting proteins bind directly to Dystroglycan's extensive glycan chains through their LG domains. Importantly, Dystroglycan can bind multiple proteins simultaneously, and increasing the length of its glycan chains increases its ligand binding capacity, suggesting it functions as a 'tunable' scaffold (*Goddeeris et al., 2013*). Dystroglycan binds Laminins to regulate the integrity of basement membranes, which can serve as a permissive growth substrate for extending axons (*Clements et al., 2017*; *Clements and Wright, 2018*; *Wright et al., 2012*). Dystroglycan also binds to the LG domain of Slits to regulate their extracellular distribution in the spinal cord (*Wright et al., 2012*). Similar to *Dystroglycan* mutants, *Slit1;Slit2, Slit1;Slit2;Slit3* and *Robo1;Robo2* mutants display defects in commissural axon crossing, as well as internal capsule, anterior commissure, and lateral olfactory tract formation (*Bagri et al., 2002*; *Fouquet et al., 2007*; *Jaworski et al., 2010*; *Long et al., 2004*; *López-Bendito et al., 2007*). However, *Slit* and *Robo* mutants do not display the prominent AP randomization seen in the commissural axons of *Ispd$^{L79*/L79*}$* and *Dag1$^{F/-}$;Sox2$^{Cre}$* mutants, raising the possibility that Dystroglycan interacts with additional molecules during axon guidance.

We therefore focused our attention on the transmembrane receptor Celsr3/Adgrc3, a mammalian orthologue of the *D. melanogaster* planar cell polarity protein (PCP) Flamingo. Celsr3 is a member of the adhesion GPCR family of proteins, and its large extracellular domain contains two LG domains, identifying it as a potential Dystroglycan interacting protein (*Figure 4A*). Celsr3 is highly expressed in the developing nervous system, including commissural and motor axons in the spinal

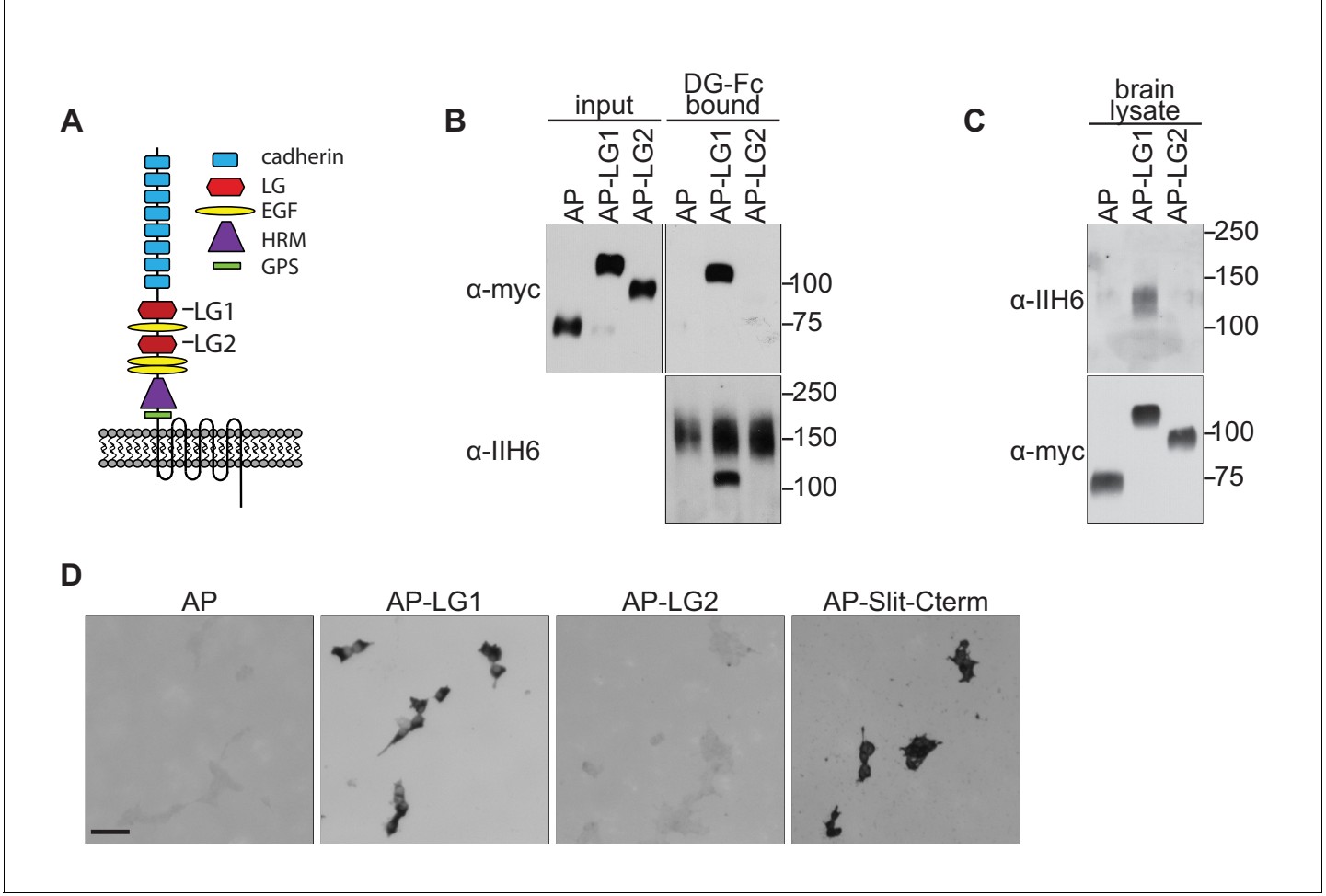

**Figure 4.** Dystroglycan interacts with the LG1 domain of Celsr3. (**A**) Schematic of Celsr3 protein structure, highlighting the location of Cadherin, Laminin G (LG), EGF, Hormone Receptor Domain (HRM) and GPCR Proteolytic Site (GPS) domains. (**B**) Fc-tagged α-Dystroglycan (Fc-DG) secreted from 293 T cells was incubated with Alkaline Phosphatase (AP)-tagged Celsr3-LG1, Celsr3-LG2, or AP-tag alone, and complexes were isolated with Protein A/G beads. DG-Fc interacts selectively with Celsr3-LG1, but not Celsr3-LG2. (**C**) AP-Celsr3-L1, AP-Celsr3-LG2, or AP-tag alone were incubated with WGA enriched brain lysate, and complexes were purified with Ni-NTA beads. AP-Celsr3-LG1 binds endogenous glycosylated Dystroglycan, whereas AP-Celsr3-LG2 and AP-tag do not. (**D**) COS7 cells transfected with full-length Dystroglycan were incubated with 5 nM AP-tag, AP-Celsr3-LG1, AP-Celsr3-LG2, or AP-Slit-Cterm. Both AP-Celsr3-LG1 and AP-Slit-Cterm exhibited selective binding. Scale bar = 50 μm.

DOI: https://doi.org/10.7554/eLife.42143.009

cord, and throughout the developing forebrain, including CTAs, TCAs, and Isl1 +guidepost cells in the ventral telencephalon (*Chai et al., 2014*; *Feng et al., 2016*; *Onishi et al., 2013*; *Zhou et al., 2008*). *Celsr3*$^{-/-}$ mutants also show remarkably similar axon guidance defects to *Ispd*$^{L79*/L79*}$ and *Dag1*$^{F/-}$;*Sox2*$^{Cre}$ mutants, exhibiting AP randomization of post-crossing commissural axons in the spinal cord, as well as defects in anterior commissure and internal capsule formation in the forebrain (*Onishi et al., 2013*; *Tissir et al., 2005*; *Zhou et al., 2008*).

To test for interactions between Dystroglycan and Celsr3, we initially attempted to generate a secreted, tagged full-length extracellular domain of Celsr3, but were unable to obtain sufficient quantities for binding studies. Similar problems occurred when we removed the eight cadherin domains and attempted to generate a version containing the three EGF and two Laminin G domains. We therefore tested whether Dystroglycan could bind to the isolated LG domains of Celsr3. We found that Fc-tagged Dystroglycan bound specifically to Alkaline Phosphatase-tagged Celsr3-LG1 domain (AP-Celsr3-LG1), but surprisingly not the AP-Celsr3-LG2 domain (*Figure 4B*). Similarly, using tagged Celsr3-LG domains as bait, we found that AP-Celsr3-LG1, but not AP-Celsr3-LG2, was able to bind endogenous glycosylated Dystroglycan from brain lysate (*Figure 4C*). In a live cell binding

assay, AP-Celsr3-LG1 and AP-Slit-C-terminal domains bound to COS7 cells overexpressing Dystroglycan, whereas AP-Celsr3-LG2 did not (*Figure 4D*). These results identify Celsr3, via its LG1 domain, as a novel binding partner for Dystroglycan.

We next sought to better understand why Dystroglycan binds to Celsr3-LG1, but not Celsr3-LG2. Recent crystal structures have provided insight into how the glycan chains of Dystroglycan bind specifically to LG domains (*Briggs et al., 2016*). GlcA-Xyl repeats (matriglycan) on Dystroglycan bind a groove in the Laminin-$\alpha$2-LG4 domain that contains a $Ca^{2+}$ binding site surrounded by several basic residues and a glycine at the tip of the loop (*Figure 5A*). These residues are all present in the LG domains of the known Dystroglycan binding proteins Laminin-$\alpha$1, Agrin, Perlecan, Pikachurin, Neurexin, and Slit, suggesting they represent a conserved binding motif between LG domains and the glycan chains on Dystroglycan. Alignment of Celsr3-LG1 with Laminin-$\alpha$2-LG4 shows significant sequence similarity, including the conservation of the basic residues, the $Ca^{2+}$ binding site, and the glycine at the end of the loop (*Figure 5A*). This region of Celsr3-LG1 is also evolutionarily conserved (*Figure 6—figure supplement 1A*). In contrast to Celsr3-LG1, Celsr3-LG2 lacks a $Ca^{2+}$ binding site, the basic residues, and the glycine, and exhibits no sequence conservation with other Dystroglycan-binding LG domains (data not shown), likely explaining its lack of binding.

To test whether this conserved region of Celsr3-LG1 was required for Dystroglycan binding, we generated GFP-tagged Celsr3 with a mutation at position 1548 (Celsr3$^{R1548Q}$-GFP). This residue corresponds to R2803 in the LG4 domain of Laminin-$\alpha$2, and is required for its binding to glycosylated Dystroglycan (*Wizemann et al., 2003*). Compared to wild-type Celsr3-GFP, in which the C-terminal 346 amino acids of the intracellular domain of Celsr3 were replaced with the coding sequence for EGFP, Celsr3$^{R1548Q}$-GFP showed similar subcellular localization and both total and cell-surface expression in 293 cells, suggesting that the R1548Q mutation does not affect the folding or stability of Celsr3 (*Figure 5B–C*). We next investigated how mutating this residue in the isolated LG1 domain (AP-Celsr3-LG1$^{R1548Q}$) would affect binding to Dystroglycan. Compared to wild-type AP-Celsr3-LG1, AP-Celsr3-LG1$^{R1548Q}$ exhibited markedly reduced binding to DG-Fc, indicating that the conserved binding interface is critical for the specificity of this interaction (*Figure 5D*).

To test the specificity of Celsr3-LG1 binding to Dystroglycan in vivo, we utilized an AP-section binding assay. E12.5 spinal cord sections and E14.5 brain sections were incubated with AP alone, AP-Celsr3-LG1, or AP-Celsr3-LG1$^{R1548Q}$. In E12.5 spinal cord, AP-Celsr3-LG1 binding was observed in post-crossing commissural axons in the ventrolateral funiculus (arrows, *Figure 5G*), similar to the expression pattern of Dystroglycan (*Figure 1A*, *Figure 1—figure supplement 1A*). In E14.5 brain sections, AP-LG1-Celsr3 binding was observed on axons in the internal capsule (arrows, *Figure 5H*), similar to the expression pattern of Dystroglycan (*Figure 2—figure supplement 1A,B*). The specificity of this binding was confirmed by the lack of AP alone binding in either the spinal cord or brain sections (*Figure 5E,F*). Furthermore, AP-Celsr3-LG1$^{R1548Q}$ showed diminished binding in both the spinal cord and brain (*Figure 5I,J*), consistent with our in vitro binding results (*Figure 5D*) and confirming the binding specificity between Dystroglycan and Celsr3-LG1.

## Dystroglycan:Celsr3 interactions are specifically required for anterior turning of commissural axons

The axon guidance phenotypes we observed in *Dystroglycan* and *Ispd* mutants are similar to those seen in *Slit/Robo* and *Celsr3* mutants. However, because Dystroglycan binds multiple LG-domain containing proteins through its glycan chains, the phenotypes identified in *Dystroglycan* and *Ispd* mutants likely reflect interactions with multiple extracellular proteins, including Laminins, Slits and Celsr3. To define which aspects of Dystroglycan-dependent axon tract formation require interactions with Celsr3, we used CRISPR/Cas9 genome editing to generate a knock-in mouse carrying an arginine-to-glutamine mutation at position 1548 in Celsr3 (*Celsr3$^{R1548Q}$*). *Celsr3$^{R1548Q/R1548Q}$* mice are viable and fertile, as opposed to *Celsr3$^{-/-}$* mice, which die immediately after birth due to respiratory defects (*Tissir et al., 2005*). Analysis of brain lysates indicated that Celsr3 protein in *Celsr3$^{R1548Q/R1548Q}$* mice migrates at the correct molecular weight, is present at normal levels, and does not lead to compensatory changes in the levels of Celsr1 (*Figure 6A*).

We first examined spinal commissural axon crossing and anterior turning in *Celsr3$^{R1548Q/R1548Q}$* mutants in open-book preparations. Remarkably, post-crossing commissural axons exhibited randomization along the AP axis, similar to *Celsr3$^{-/-}$*, *Ispd$^{L79*/L79*}$*, and *Dag1$^{F/-}$;Sox2$^{Cre}$* mutants (*Figure 6C*). Quantification shows that only 22.32 ± 6.35% of injection sites in *Celsr3$^{R1548Q/R1548Q}$*

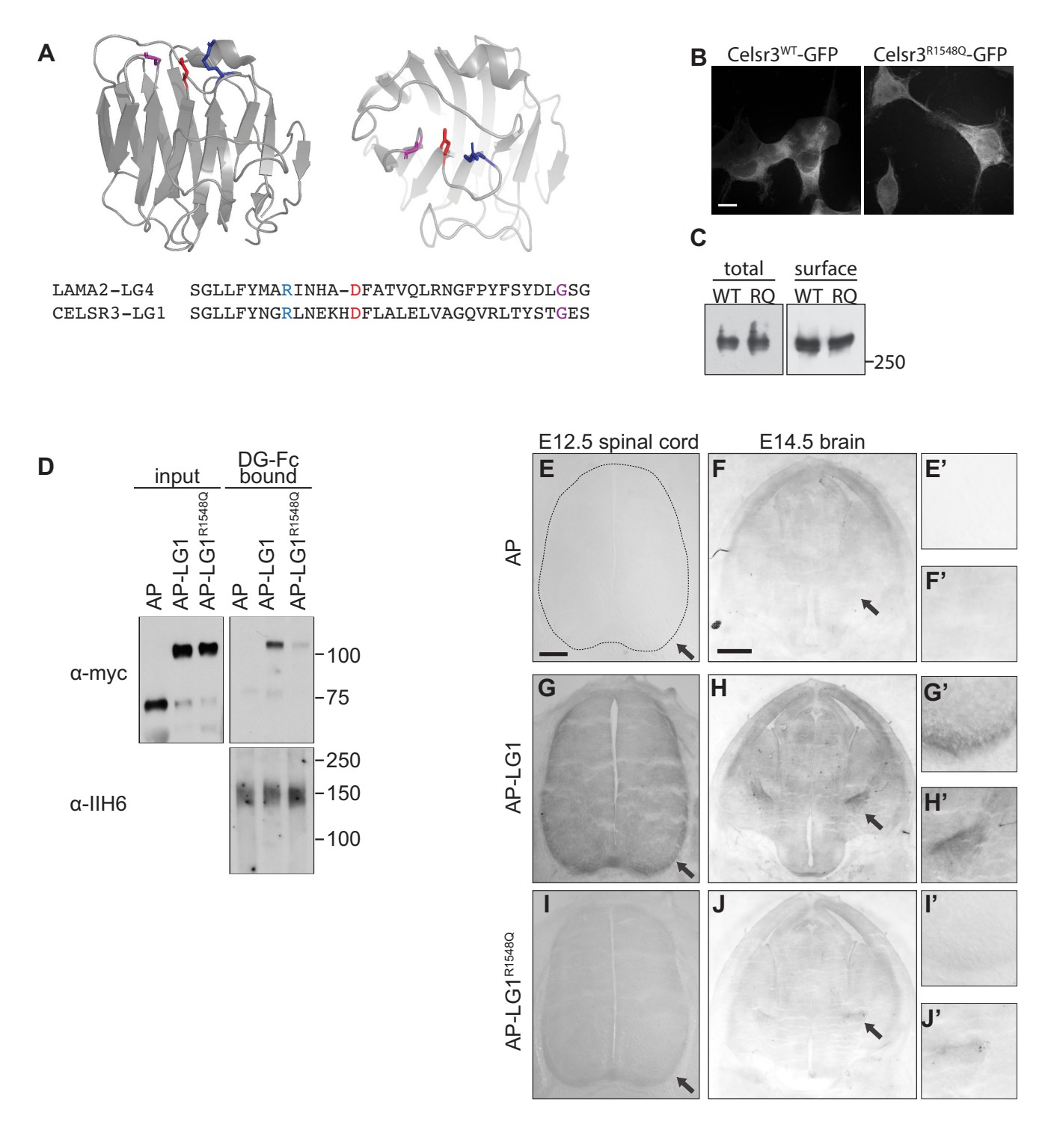

**Figure 5.** Dystroglycan binding requires specific motifs in Celsr3 LG1. (**A**) Top: schematic showing the structure of the LG4 domain of Laminin-α2 (PDB:1OKQ), highlighting conserved residues critical for Dystroglycan binding: Arginine2803 (blue), Aspartate2808 (red) and Glycine2826 (purple). Bottom: Partial sequence alignment of murine Celsr3-LG1 (amino acids 1540–1574) with murine Laminin-α2-LG4 (amino acids 2795–2828) shows conservation at Arginine1548 (blue), Aspartate1564 (red) and Glycine1572 (purple) of Celsr3. (**B–C**) 293 T cells transfected with Celsr3-GFP or mutant Celsr3R1548Q-GFP showed no differences in expression levels or cell surface localization by immunocytochemistry (**B**) or western blotting (**C**). (**D**) Mutation of Celsr3-LG1 at Arginine1548 (AP-LG1R1548Q) results in loss of binding to FC-tagged Dystroglycan. (**E–J**) Section binding assay with 5 nM AP

*Figure 5 continued on next page*

Figure 5 continued

alone (E–F), AP-Celsr3-LG1 (G–H), or AP-Celsr3-LG1$^{R1548Q}$ (I–J). Inset panels show higher magnification of the ventrolateral funiculus (top panels E', (G', I') and the internal capsule (bottom panels, (F', H', J'). AP-Celsr3-LG1 binds to commissural axons in the ventrolateral funiculus (arrow, G,G') and the internal capsule (arrow, H,H'). AP-Celsr3-LG1$^{R1548Q}$ shows minimal binding in either region and is almost indistinguishable from AP alone. Scale bar = 10 µm (B), 100 µm (E,G,I), 500 µm (F,H,J).

DOI: https://doi.org/10.7554/eLife.42143.010

mutants had normal anterior turning, while the remaining 77.68% exhibited AP randomization. This phenotype was fully penetrant, with all *Celsr3$^{R1548Q/R1548Q}$* mutants showing abnormal turning at multiple injection sites. Importantly, none of the *Celsr3$^{R1548Q/R1548Q}$* mutants exhibit the floorplate stalling phenotypes that are seen in *Ispd$^{L79*}$*, *Dag1$^{F/-}$;Sox2$^{Cre}$*, and *Slit/Robo* compound mutants, consistent with Celsr3's specific role in regulating anterior turning in post-crossing commissural axons. E12.5 spinal cord sections labeled with antibodies to L1 or Robo1 and Robo2 (*Figure 6—figure supplement 1A–E*) confirm that the overall structure of the ventrolateral funiculus in *Celsr3$^{R1548Q/R1548Q}$* mutants appears normal. This is consistent with *Celsr3$^{F/F}$;Wnt1$^{Cre}$* mutants, which also exhibit AP randomization but normal L1 staining of post-crossing axons, indicating that despite their AP randomization, these axons still extend within the ventrolateral funiculus (*Onishi et al., 2013*). Overall, these results show that Celsr3 interacts with Dystroglycan through its LG1 domain to direct the proper anterior turning of post-crossing commissural axons.

In contrast to the results we observed in the spinal cord, the internal capsule and other axon tracts in the forebrains of *Celsr3$^{R1548Q/R1548Q}$* mutants appeared normal by both immunostaining and DiI labeling (*Figure 6E–J*). Therefore, the in vivo requirement for Dystroglycan:Celsr3 interactions appears to be context dependent, and the defects in forebrain axon tract formation in *Dystroglycan* and *Ispd$^{L79*/L79*}$* mutants likely reflect Dystroglycan interactions with other LG-domain containing proteins such as Laminins or Slits.

## Discussion

The neurological abnormalities in patients with dystroglycanopathy are extremely heterogeneous, ranging from mild cognitive defects to severe and widespread structural abnormalities (*Godfrey et al., 2011*). Severe forms of Dystroglycanopathy (WWS, MEB) are characterized by profound neurodevelopmental defects that can include type II lissencephaly, hydrocephalus, hindbrain hypoplasia, and defects in white matter. Interestingly, congenital mirror movements, which arise from improper decussation of descending corticospinal axons as they pass through the brainstem, have been reported in isolated cases of dystroglycanopathy, suggesting that axon tract abnormalities may contribute to the neuropathology of this disorder (*Ardicli et al., 2017*; *Longman et al., 2003*).

Using a model of severe dystroglycanopathy (*Ispd$^{L79*}$*) and *Dystroglycan* conditional mutants, we now show that Dystroglycan is required for proper development of several major axon tracts, including commissural axons in the spinal cord and several major axon tracts in the forebrain. Taken with our previous results, these findings demonstrate that axon guidance defects are a key feature of dystroglycanopathy, which arise due to Dystroglycan's interaction with multiple ECM proteins, secreted axon guidance cues, and transmembrane axon guidance receptors.

### Dystroglycan interacts with multiple LG domain containing proteins to regulate axon guidance at intermediate targets

As axons develop, guidepost cells function as intermediate targets and express molecular cues that direct them to their final targets in a step-wise manner (*Squarzoni et al., 2015*). These guidepost cells include glia, neurons, and other axons. In the developing spinal cord, commissural axons are initially directed ventrally by repulsive cues emanating from specialized cells in the roof plate, and extend along the basal endfeet of neuroepithelial cells, where Netrin accumulates to generate a permissive substrate for commissural axon growth (*Augsburger et al., 1999*; *Butler and Dodd, 2003*; *Varadarajan and Butler, 2017*; *Varadarajan et al., 2017*). These axons then encounter a specialized population of midline glial cells at the floor plate that express a number of other attractive and repulsive guidance cues, including Netrin, VEGF, Shh, Slits, and Semaphorins that promote crossing

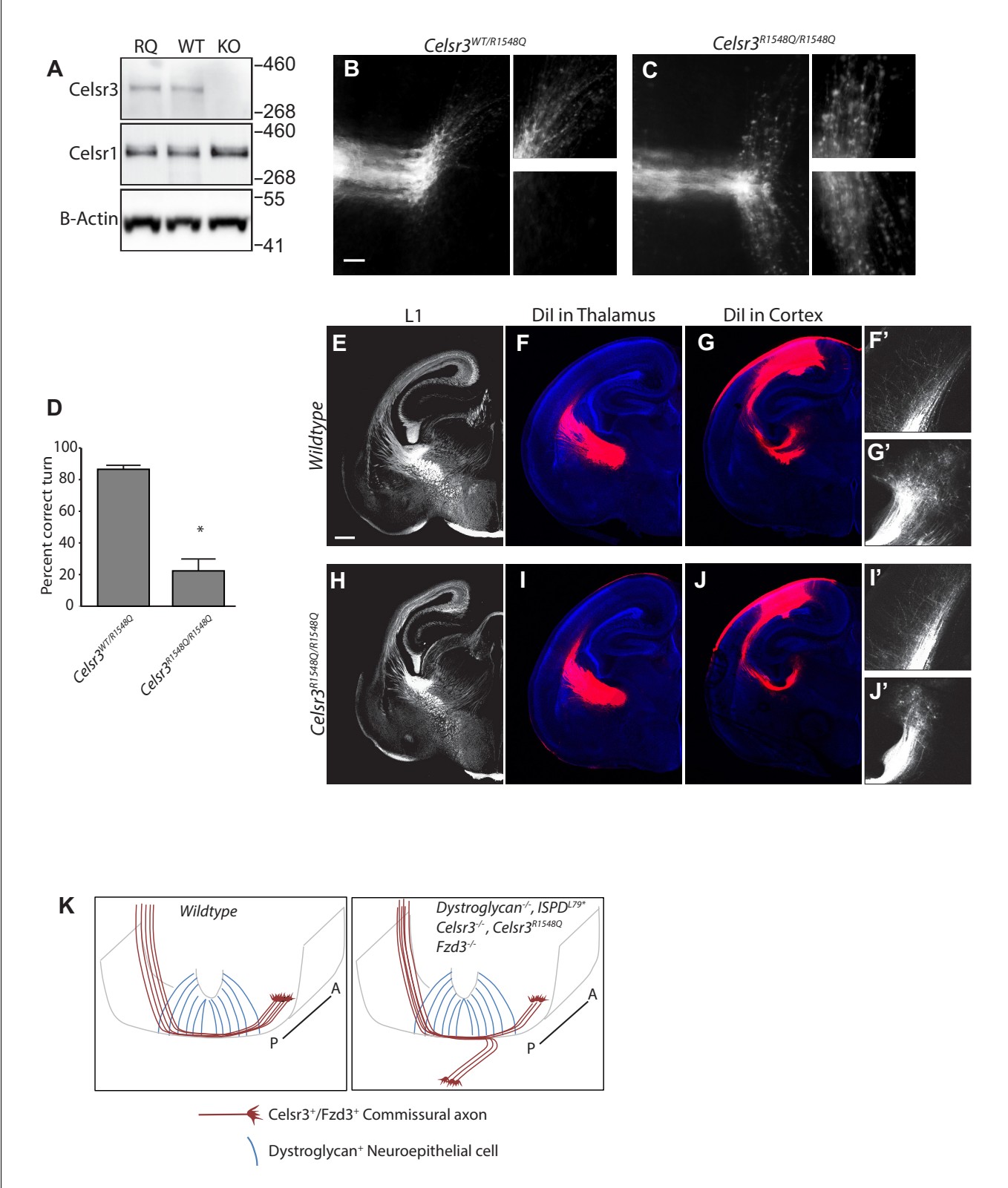

**Figure 6.** Dystroglycan:Celsr3 interactions are required for spinal commissural axon guidance. (A) Western blotting of brain lysates from *Celsr$^{R1548Q/R1548Q}$* mutants and wildtype littermates show no difference in size or expression level of Celsr3 or Celsr1 protein. Brain lysate from *Celsr3$^{-/-}$* mutants is included as a control for antibody specificity. (B, C). In *Celsr3$^{WT/R1548Q}$* heterozygous controls (B), DiI labeling of open book preparations shows that commissural axons extend through the floor plate, then execute an anterior turn in 86.5 ± 2.52% of injection sites (n = 7 animals, 49 total injection sites).
*Figure 6 continued on next page*

*Figure 6 continued*

In contrast, only 22.32 ± 6.35% of injection sites in *Celsr3*[R1548Q/R1548Q] mutants (n = 6 animals, 48 total injection sites) (**C**) show normal anterior turning, with the remaining 77.68% exhibiting AP randomization after crossing the floor plate, similar to *Dag1*[F/-];*Sox2*[Cre], *Ispd*[L79*/L79*], and *Celsr3*[-/-] mice. Higher magnification insets for each image show the anterior (top) and posterior (bottom) trajectories of post-crossing commissural axons. (**D**) Quantification of open book preparations, *p<0.001, Student's T-test. (**E–J**) L1 immunohistochemistry (**E, H**) and DiI labeling of thalamocortical (**F, I**) and corticothalamic (**G, J**) axons show no defects in internal capsule formation in *Celsr3*[R1548Q] mutants. High magnification insets show DiI-labeled thalamocortical axons extending into the intermediate zone of the cortex (**F', I'**) and DiI-labeled corticothalamic axons entering the thalamus (**G',J'**). (**K**) Proposed model for Dystroglycan:Celsr3 interactions in guiding commissural axons. Scale bar = 50 μm (**B,C**), 500 μm (**E–J**).
DOI: https://doi.org/10.7554/eLife.42143.011

The following figure supplement is available for figure 6:

**Figure supplement 1.** Analysis of *Celsr*[R1548Q] mutants.
DOI: https://doi.org/10.7554/eLife.42143.012

at the ventral midline and prevent recrossing (*Charron et al., 2003*; *Kennedy et al., 1994*; *Long et al., 2004*; *Nawabi et al., 2010*; *Ruiz de Almodovar et al., 2011*). In addition, cell adhesion molecules expressed on both commissural axons and within the floor plate are critical for proper midline crossing (*Stoeckli and Landmesser, 1995*; *Stoeckli et al., 1997*). Finally, an anterior[high]:posterior[low] gradient of Wnts guides commissural axons rostrally towards the brain (*Lyuksyutova et al., 2003*). Similar to the spinal cord, cues derived from specialized midline glia in the brain guide axons across the dorsal midline through the corpus collosum and the anterior commissure and optic chiasm in the ventral forebrain (*Bagri et al., 2002*; *Erskine et al., 2011*; *Marcus et al., 1995*; *Shu and Richards, 2001*; *Silver and Ogawa, 1983*; *Williams et al., 2003*).

Our data showing that Dystroglycan functions non-cell autonomously and does not require signaling through its intracellular domain to regulate axon tract formation suggests that Dystroglycan primarily functions at intermediate targets as an extracellular scaffold. Combined with our previous results, we have now identified three mechanisms by which Dystroglycan regulates axon guidance. We previously showed that glycosylated Dystroglycan is required for the organization of Laminins and other ECM proteins in the spinal cord basement membrane. This provides a permissive growth substrate for axons as they cross the ventral midline in the spinal cord and forebrain and extend towards their final targets (*Clements and Wright, 2018*; *Wright et al., 2012*). In addition, Dystroglycan binds to the LG domain in the secreted axon guidance cues Slits, ensuring their proper extracellular localization within the spinal cord floorplate (*Wright et al., 2012*). In this current study, we now identify a novel interaction between Dystroglycan and the transmembrane axon guidance receptor Celsr3 that is required for the proper anterior turning of post-crossing commissural axons. Remarkably, mutating a key residue in the LG1 domain of Celsr3 (*Celsr3*[R1548Q]) disrupts its binding to Dystroglycan in vitro and results in AP randomization of post-crossing spinal commissural axons in vivo, similar to *Ispd*[L79*/L79*], *Dag1*[F/-];*Sox2*[Cre], and *Celsr3*[-/-] mutants. Therefore, we propose a model in which Celsr3, functioning cell-autonomously in the growth cones of commissural axons (*Onishi et al., 2013*), binds in trans to Dystroglycan present in the neuroepithelium and/or basement membrane as axons cross the floorplate (*Figure 6F*). Previous work has shown that Celsr3 and Fzd3 are required to direct the anterior turning of commissural axons in response to a Wnt gradient in the floor plate. How this occurs is still not well understood, but it involves downstream signaling pathways involving Jnk and atypical PKC (*Lyuksyutova et al., 2003*; *Onishi et al., 2013*; *Onishi and Zou, 2017*; *Wolf et al., 2008*).

We also find that Dystroglycan is critical for proper formation of several axon tracts in the developing forebrain, including the internal capsule, the anterior commissure, and the lateral olfactory tract. These axon tracts all require molecular cues from guidepost cells at intermediate targets to develop properly. During internal capsule formation, TCAs are directed laterally by cues from the ventral diencephalon, then grow across the DTB along an 'axon bridge' comprised of projections from Isl + neurons in the prethalamus and ventral telencephalon (*Braisted et al., 1999*; *Feng et al., 2016*). TCAs are then directed through the ventral telencephalon by a population of LGE-derived Isl1 +guidepost neurons that form a permissive 'corridor', and require interactions with descending CTAs to cross the PSPB (*López-Bendito et al., 2006*; *Squarzoni et al., 2015*). Once in the cortex, TCAs turn medially and are initially confined to the intermediate zone by cortical subplate neurons for a several day 'waiting period' prior to their extension into layer 4 (*Ghosh et al., 1990*). During

anterior commissure development, Nkx2.1+glia function as guidepost cells at the ventral midline (*Minocha et al., 2015*). Guidance of axons in the lateral olfactory tract requires a subset of Cajal-Retzius guidepost neurons in the ventrolateral telencephalon (*Dixit et al., 2014*; *Sato et al., 1998*).

Dystroglycan and Ceslr3 mutants have similar phenotypes in all three of these forebrain axon tracts. Surprisingly, however, forebrain axon tract development was normal in *Celsr3$^{R1548Q}$* mutants, suggesting that Dystroglycan:Celsr3 interactions are not required in the forebrain. How might Celsr3 be functioning independently of Dystroglycan during forebrain axon guidance? Celsr3 is required in the Isl1$^+$ guidepost neurons in the prethalamus and ventral telencephalon that form the 'axon bridge' that is required for TCAs to cross the DTB (*Feng et al., 2016*). Therefore, it is possible that the cadherin repeats of Celsr3 mediate homophilic interactions between the Isl1 +axons, but this has not been tested directly. Celsr3 is also required cell autonomously for peripheral extension of motor axons into the limb, where they form a complex with Ephrin-A2, -A5, and Ret (*Chai et al., 2014*). In contrast, *Dystroglycan* and *Celsr3$^{R1548Q}$* mutants do not display any defects in peripheral motor axon growth (data not shown). Therefore, the importance of Dystroglycan:Celsr3 interactions during axon guidance are context dependent.

If Dystroglycan:Celsr3 interactions are not required for internal capsule formation, what could explain the severe forebrain axon guidance phenotypes in *Ispd$^{L79*/L79*}$* and *Dag1$^{F/-}$;Sox2$^{Cre}$* mutants? Dystroglycan has the ability to bind multiple LG-domain containing partners simultaneously through its extensive glycan chains, making it difficult to ascribe function to a single molecular interaction. However, Slits are likely candidates in the forebrain, as *Slit* and *Robo* mutants exhibit defects in internal capsule, anterior commissure and lateral olfactory tract development similar to *Ispd$^{L79*/L79*}$* and *Dag1$^{F/-}$;Sox2$^{Cre}$* mutants (*Bagri et al., 2002*; *Bielle et al., 2011*; *Fouquet et al., 2007*; *López-Bendito et al., 2007*; *Nguyen-Ba-Charvet et al., 2002*). TCAs, CTAs, and olfactory axons are all responsive to Slit, and Slit expression by Nkx2.1+midline glia regulates anterior commissure formation (*Braisted et al., 2009*; *Li et al., 1999*; *Minocha et al., 2015*; *Nguyen Ba-Charvet et al., 1999*; *Shu and Richards, 2001*). In addition to directly repelling axons, Slits also regulate the migration of guidepost neurons in the internal capsule and lateral olfactory tract, suggesting the axon guidance phenotypes may be secondary to neuronal migration defects (*Bielle et al., 2011*; *Fouquet et al., 2007*). Our results demonstrate that Dystroglycan functions non-cell autonomously in neuroepithelial cells, and not in axons or guidepost cells, during forebrain axon tract development. Therefore, Dystroglycan may influence neuronal migration and axon guidance in the forebrain by regulating the distribution of Slit proteins, similar to its role in the ventral midline of the spinal cord. Determining precisely how these pathways interact to regulate axon tract formation will require careful spatial and temporal manipulation of their expression in vivo.

## Evolutionary conservation of dystroglycan function during axon guidance

Dystroglycan and its binding partners have evolutionarily conserved roles in regulating axon guidance. The *C. elegans* Dystroglycan homologue DGN-1 is required for follower axons to faithfully track along pioneer axons (*Johnson and Kramer, 2012*). Similarly, FMI-1, the *C. elegans* homologue of Celsr3, is involved in both pioneer and follower axon guidance in the ventral nerve cord (*Steimel et al., 2010*). FMI-1 phenotypes could be rescued by expressing the regions encompassing either the cadherin repeats or the EGF and LG domains of FMI-1, suggesting that it may function in both a homophilic and heterophilic manner, depending on the context. In *D. melanogaster*, the Dystroglycan homologue Dg functions in both neurons and glial cells to guide the proper targeting of photoreceptor axons to the optic lobe (*Shcherbata et al., 2007*). *Slit* mutants and RNAi to *Robo1/2/3* show a remarkably similar photoreceptor targeting phenotype to *Dg* mutants, which arises from a failure to form a proper boundary between the lamina glia and the lobula cortex (*Tayler et al., 2004*). *Flamingo*, the Drosophila homologue of Celsr3, functions at a subsequent step in visual system development to non-cell autonomously regulate synaptic choice of photoreceptors (*Chen and Clandinin, 2008*; *Lee et al., 2003*; *Senti et al., 2003*). The remarkable similarities in axon targeting defects seen in *Dystroglycan*, *Slit* and *Celsr3* mutants across species suggests that their interactions are evolutionarily conserved.

## Dystroglycan regulates several aspects of nervous system development by binding to multiple proteins

In addition to regulating axon guidance decisions throughout the nervous system, Dystroglycan is required for neuronal migration, synapse formation, glial development, and maintenance of the blood-brain barrier (*Clements et al., 2017*; *Früh et al., 2016*; *McClenahan et al., 2016*; *Menezes et al., 2014*; *Michele et al., 2002*; *Moore et al., 2002*; *Myshrall et al., 2012*; *Saito et al., 2003*; *Satz et al., 2008*; *Satz et al., 2010*; *Wright et al., 2012*). The widespread nature of these defects reflects the reiterative function of Dystroglycan throughout neurodevelopment and its interactions with multiple partners.

During early neurodevelopment, Dystroglycan maintains the attachment of neuroepithelial cells to the basement membrane in the brain and retina. These neuroepithelial cells serve as scaffolds for neuronal migration. Loss of Dystroglycan in neuroepithelial cells results in type II lissencephaly in the cortex, cerebellar migration defects, and ectopic migration of inner retinal neurons into the vitreous of the retina (*Clements et al., 2017*; *Nguyen et al., 2013*; *Satz et al., 2010*). In contrast, neuronal migration is unaffected by the deletion of Dystroglycan from postmitotic neurons themselves. Laminins and Perlecan are likely the primary binding partners for neuroepithelial Dystroglycan in this context, as they are highly enriched in basement membranes, their localization to basement membranes is disrupted in the absence of Dystroglycan, and both Laminin and Perlecan mutants exhibit similar neuronal migrations defects (*Costell et al., 1999*; *Edwards et al., 2010*; *Halfter et al., 2002*; *Ichikawa-Tomikawa et al., 2012*; *Pinzón-Duarte et al., 2010*; *Radmanesh et al., 2013*).

At later stages of neurodevelopment, Dystroglycan is expressed in neurons, where it regulates specific subsets of perisomatic inhibitory synapses and hippocampal LTP (*Früh et al., 2016*; *Satz et al., 2010*; *Zaccaria et al., 2001*). It is unclear which proteins Dystroglycan interacts with at synapses, although Neurexins are possible candidates, as they have been shown to bind to Dystroglycan through their LG domains (*Reissner et al., 2014*; *Sugita et al., 2001*). Several other LG-domain containing proteins with the conserved Dystroglycan binding motif (Celsrs, CNTNAPs, Thrombospondins, Laminins) are also localized to synapses, suggesting that Dystroglycan may have a complex and context specific role in synapse formation and maintenance. Importantly, patients with milder forms of dystroglycanopathy can have cognitive defects even in the absence of any obvious structural abnormalities in the brain, which may reflect the role of synaptic Dystroglycan.

In light of Dystroglycan's binding to multiple ligands important for nervous system development, it is interesting to note that Dystroglycan displays differential glycosylation patterns in muscle, glia, and even between neuronal subtypes. By western blotting, glial Dystroglycan migrates at ~120 kD, Dystroglycan in cortical/hippocampal neurons migrates slightly higher (~140 kD), whereas Dystroglycan in cerebellar Purkinje neurons migrates at ~180 kD (*Satz et al., 2010*). How these differences in glycosylation arise in distinct cell types remains unclear, but the observation that the length of the glycan chains correlates with binding capacity suggests that it is likely to have functional consequences, particularly in the nervous system.

In summary, our results establish a widespread role for Dystroglycan in regulating axon tract formation throughout the developing nervous system. We also identify Celsr3 as a novel binding partner for Dystroglycan and find that their interaction is required for anterior turning of post-crossing commissural axons. By functioning as an extracellular scaffold that binds multiple ECM proteins, secreted axon guidance cues, and transmembrane receptors, Dystroglycan plays a critical role in many aspects of neural circuit development and function.

## Materials and methods

**Key resources table**

| Reagent type (species) or resource | Designation | Source or reference | Identifiers | Additional information |
|---|---|---|---|---|
| Genetic reagent (M. musculus) | Dag1[Flox] | Jackson Labs | stock: 009652 | |

*Continued on next page*

*Continued*

| Reagent type (species) or resource | Designation | Source or reference | Identifiers | Additional information |
|---|---|---|---|---|
| Genetic reagent (M. musculus) | Dag$^{Bcyto}$ | PMID: 19846701 | | Dr. Kevin Campbell, HHMI, University of Iowa |
| Genetic reagent (M. musculus) | Ispd$^{L79*}$ | PMID: 23217742, Jackson Labs | stock: 022019 | Dr. Kevin Wright, Vollum Institute |
| Genetic reagent (M. musculus) | Celsr3$^{R1548Q}$ | generated de novo | | Dr. Kevin Wright, Vollum Institute |
| Genetic reagent (M. musculus) | Sox2Cre | Jackson Labs | stock: 008454 | |
| Genetic reagent (M. musculus) | Wnt1Cre | Jackson Labs | stock: 022137 | |
| Genetic reagent (M. musculus) | Foxg1Cre | Jackson Labs | stock: 006084 | |
| Genetic reagent (M. musculus) | Gbx2CreERT2 | Jackson Labs | stock: 022135 | |
| Genetic reagent (M. musculus) | Emx1Cre | Jackson Labs | stock: 005628 | |
| Genetic reagent (M. musculus) | Dlx5/6Cre | Jackson Labs | stock: 008199 | |
| Genetic reagent (M. musculus) | R26-LSL-TdTomato | Jackson Labs | stock: 007909 | |
| Cell line (H. sapiens) | 293T | ATCC | CRL-11268, RRID:CVCL_1926 | |
| Cell line (C. aethiops) | COS7 | ATCC | CRL-1651, RRID:CVCL_0224 | |
| Antibody | L1 (rat monoclonal) | Millipore | RRID: AB_2133200 | 1:500 dilution |
| Antibody | Robo1 (goat polyclonal) | R and D Systems | RRID: AB_354969 | 1:250 dillution |
| Antibody | Robo2 (goat polyclonal) | R and D Systems | RRID: AB_2181857 | 1:250 dilution |
| Antibody | Dystroglycan (rabbit polyclonal) | Santa Cruz Biotech | RRID: AB_1118902 | 1:50 dilution |
| Antibody | myc (mouse monoclonal) | Thermo Fisher | RRID: AB_2533008 | |
| Antibody | IIH6 glcosylated dystroglycan (mouse monoclonal) | Millipore | RRID: AB_309828 | |
| Antibody | Celsr3 (rabbit poyclonal) | Fadel Tissir | | Dr. Fadel Tissir, UC Louvain |
| Antibody | Celsr1 (guinea pig polyclonal) | Fadel Tissir | | Dr. Fadel Tissir, UC Louvain |
| Transfected construct (O. cuniculus) | DG-Fc | PMID: 11604425 | | Dr. Kevin Campbell, HHMI, University of Iowa |
| Transfected construct (M. musculus) | Celsr3-GFP | PMID: 25108913 | | Dr. Fadel Tissir, UC Louvain |

*Continued on next page*

*Continued*

| Reagent type (species) or resource | Designation | Source or reference | Identifiers | Additional information |
|---|---|---|---|---|
| Transfected construct (M. musculus) | Celsr3-GFP-R1548Q | generated de novo | | Dr. Kevin Wright, Vollum Institute |
| Transfected construct (synthetic vector) | AP-Tag5-COMP | generated de novo | | Dr. Kevin Wright, Vollum Institute |
| Transfected construct (M. musculus) | AP-Tag5-COMP -Celsr3-LG1 | generated de novo | | Dr. Kevin Wright, Vollum Institute |
| Transfected construct (M. musculus) | AP-Tag5-COMP -Celsr3-LG2 | generated de novo | | Dr. Kevin Wright, Vollum Institute |
| Transfected construct (M. musculus) | AP-Tag5-COMP- Celsr3-LG1-R1548Q | generated de novo | | Dr. Kevin Wright, Vollum Institute |
| Chemical compound, drug | DiI | Thermo Fisher | D-3911 | |
| Chemical compound, drug | BCIP | Roche Applied Science | Cat: 11383221001 | |
| Chemical compound, drug | NBT | Roche Applied Science | Cat: 11383213001 | |
| Commercial assay or kit | cell surface biotinylation kit | Thermo Fisher | Cat: 89881 | |

## Generation and analysis of mutant mice

*Ispd$^{L79*}$* (**Wright et al., 2012**), *Dystroglycan$^{Flox/Flox}$* (**Michele et al., 2002**), *Dystroglycan$^{βcyto}$* (**Satz et al., 2009**), *Sox2$^{Cre}$* (**Hayashi et al., 2002**), *Foxg1$^{Cre}$* (**Hébert and McConnell, 2000**), *Gbx2$^{CreERT2}$* (**Chen et al., 2009**) *Emx1$^{Cre}$* (**Gorski et al., 2002**), *Dlx5/6$^{Cre}$* (**Stenman et al., 2003**), and *Wnt1$^{Cre}$* mice were maintained on a C57Bl/6J background. *Ai9/R26$^{LSL-TdTomato}$* mice were maintained on an outbred CD1 background.

*Celsr3$^{R1548Q}$* mice were generated using CRISPR/Cas9 pronuclear injection by the HHMI/Janelia Farm Gene Targeting and Transgenic Facility. The gRNA (5'-AAAAGTCATGCTTCTCGTTC-3') was co-injected with 163 bp ssDNA (5' -ATGTCTGATCCTAATGGTCCCACTCCACTTCACTCAGG TTTGCAACTGTGCAACCCAGCGGGCTACTCTTCTACAACGGGCAGCTGAACGAGAAGCATGAC TTTTTGGCTCTAGAGCTTGTGGCTGGCCAAGTGCGGCTTACATATTCCACGGGTGGGTGCTC-3') and Cas9 protein with a concentration of 5:25:25 ng/ul. Eight founders from 58 pups were identified with mutations in Celsr3. Correctly targeted mutations were then confirmed by PCR, followed by Sanger sequencing. Three correctly targeted *Celsr3$^{R1548Q/+}$* founders were obtained, and after outcrossing to the outbred CD1 strain for two generations, *Celsr3$^{R1548Q/+}$* mice were intercrossed to generate *Celsr3$^{R1548Q/R1548Q}$* homozygous mutants. *Celsr3$^{R1548Q}$* mice were genotyped using the following primers: Fwd: 5'-CACTGGCATCTCCCACACTA-3' and Rev: 5'-GGGACACCTGAGAGGA TTCA-3'. PCR products were then incubated with PvuII, which cuts the CAGCTG site generated in the *Celsr3$^{R1548Q}$* mutants.

Mice were handled and bred in accordance with the Oregon Health and Science University IACUC guidelines. Embryos were obtained from timed pregnancies, with the date of plug appearance counted as e0.5. To generate *Dystroglycan* conditional knockouts, *Dystroglycan$^{+/-}$; Cre$^+$* male breeders were crossed to *Dystroglycan$^{Flox/Flox}$* females. All conditional knockout analyses used *Dystroglycan$^{F/+}$; Cre$^+$* littermates. Phenotypic analysis was conducted on at least three different offspring obtained from at least three different litters, using at least two different male breeders, without regard to sex of animals analyzed. Mice were genotyped by PCR as previously described.

## Immunohistochemistry and anterograde tract tracing

For analysis of brains, P0 mice were euthanized by decapitation, brains were removed and fixed in 4% paraformaldehyde at 4° overnight. For L1 immunostaining, brains were washed three times for 30 min each in PBS, then embedded in low melt agarose. 150 µm thick vibrotome sections were collected and washed once in PBS, blocked for 30 min in PBS + 0.25% TritonX-100, 5% goat serum,

then incubated in primary antibody diluted in blocking buffer at 4° for two days. Sections were washed in PBS five times for thirty minutes each, then incubated in secondary antibody diluted in blocking buffer at room temperature, overnight. Sections were then washed five times for 1 hr each in PBS, with DAPI (1:5000) included in the second wash step. Sections were then mounted on Permafrost slides, light protected with Fluoromount-G (Southern Biotech), and imaged.

For spinal cord sections, E12.5 embryos were fixed in 2% paraformaldehyde at 4°C overnight, washed three times in PBS and incubated in 15% sucrose overnight. Tissue was embedded in OCT and 20 µm thick cryosections were mounted on Permafrost glass slides. Sections were washed twice in PBS, blocked in 5% serum and 0.25% TritonX-100 in PBS for 30 min, then incubated in primary antibody diluted in blocking solution at 4°C overnight. Slides were washed three times for 10 min in PBS, then incubated in secondary antibody diluted in blocking buffer at room temperature for two hours. Sections were then washed five times for five minutes each in PBS, with DAPI (1:5000) included in the second wash step. Sections were then light protected with Fluoromount-G (Southern Biotech), and imaged.

For anterograde tract tracing, DiI crystals were inserted into the cortex or thalamus of fixed brains, returned to 4% paraformaldehyde, and incubated at 37° for 5–7 days. Brains were then embedded in low melt agarose, 150 µm thick vibrotome sections were collected in PBS, incubated in DAPI (1:5000) for 30 min, washed once in PBS for five minutes, mounted, and imaged on a Zeiss M2 Imager equipped with ApoTome. Images were processed in Zeiss Zen Blue and Adobe Photoshop six software.

## Open book preparations

Embryos were collected at E12.5 and fixed for 30 min in 0.5% paraformaldehyde. Spinal cords were then removed, split along the roof plate, the meninges were removed, and the flattened spinal cords were fixed in 4% paraformaldehyde for four hours at room temperature. DiI crystals were then inserted along the lateral margin of the spinal cord and tissue was incubated in 4% paraformaldehyde at room temperature overnight. Open book preparations were then imaged on a Zeiss ZoomV-16 dissecting microscope at 50X magnification. Each injection site was scored blind to genotype by three lab members as to whether axons correctly executed an anterior turn. Percent correct turn was calculated by dividing the number of injection sites that turn anterior by the total number of injection sites within each spinal cord. Each animal represents a single 'N'

## Binding assays

293 T cells (ATCC, CRL-11268) were transfected with constructs encoding Fc-tagged Dystroglycan (DG-Fc), AP-tag alone, AP-Celsr3-LG1, AP-Celsr3-LG-2, AP-Celsr3-LG1$^{R1548Q}$, or AP-Slit-Cterm. After recovery, cells were maintained in OptiMEM for 48–72 hr, after which supernatant was collected, concentrated by centrifugation (Amicon, 10kD molecular weight cutoff), and exchanged to binding buffer (20 mM Hepes, pH 7.0, 150 mM NaCl, 2.5 mM CaCl$_2$). DG-Fc was coupled to Protein-A agarose beads for 6 hr at 4°, beads were washed once in binding buffer, and 5 nM of AP-tagged ligand was added and beads were incubated at 4° overnight, rocking.

For endogenous Dystroglycan binding assays, brains from postnatal day 7 (P7) mice were homogenized in a 10X volume of PBS + 1% Triton, incubated for 1 hr at 4°, rocking and insoluble material was removed by centrifugation at 3400xg. Supernatant was incubated with WGA-agarose beads at 4° overnight, then competed off the WGA-beads with 500 mM N-acetyl-D-glucosamine, followed by dialysis in binding buffer at 4° overnight. WGA-enriched lysate was then incubated with AP-tagged ligands (5 nM) pre-coupled to NiNTA beads.

For all binding experiments, beads were washed five times with binding buffer to remove unbound material. Bound proteins were eluted by boiling in 1X LDS sample buffer with 50 mM DTT for 10 min, resolved by SDS-PAGE, transferred to PVDF membranes, blocked for 60 min in 5% nonfat milk in TBS +0.1% Tween-20 (TBST), then probed with antibodies diluted in blocking buffer at 4° overnight. Membranes were washed three times for 10 min in TBST, incubated with secondary antibody diluted in blocking buffer with 5% nonfat milk, washed three times for 10 min in TBST, and developed with SuperSignal ECL Pico.

For live cell binding assays, COS7 cells (ATCC, CRL-1651) plated on poly-D-lysine were transfected with myc-tagged full-length Dystroglycan. 48 hr after transfection, cells were incubated with

AP-tagged ligand at 37° for 30 min. Cells were then washed five times in HBSS, fixed with 4% paraformaldehyde +60% acetone for 30 s, and washed five times in HBSS. Plates were then incubated at 67° for 90 min to inactivate endogenous peroxidase activity. Cells were washed twice in AP buffer (100 mM Tris, pH 9.0, 50 mM MgCl$_2$), then incubated with BCIP/NBT in AP buffer for 30–60 min until signal developed. The AP reaction was stopped by washing cells twice in HBSS +50 mM EDTA, and cells were imaged on a Zeiss ZoomV-16 dissecting microscope at 100X magnification.

For AP-section binding assays, embryos (E12.5 for spinal cord; E14.5 for brain) were lightly fixed in 2% paraformaldehyde for 4 hr, washed three times in PBS and incubated in 15% sucrose overnight. Tissue was embedded in OCT and 35 μm thick cryosections were mounted on frosted glass slides. Tissue was washed twice in binding buffer (20 mM Hepes pH 7.5, 100 mM NaCl, 5 mM CaCl$_2$), then incubated in binding buffer at 68°C for 90 min to inactivate endogenous alkaline phosphatase activity. Sections were then incubated with the indicated ligands diluted to 5 nM in binding buffer at 4° for 60 min. Slides were washed five times in binding buffer to remove non-specific binding, and bound ligand was crosslinked by briefly incubating slides in 4% paraformaldehyde for 60 s. Slides were then washed in AP buffer (100 mM Tris, pH 9.0, 50 mM MgCl$_2$), then incubated with BCIP/NBT in AP buffer until signal developed. The reaction was then stopped by incubating in 4% paraformaldehyde with 50 mM EDTA for 10 min, slides were coverslipped and imaged on a Zeiss AxioZoon.V16 dissecting microscope.

## Celsr3$^{R1548Q}$-GFP generation and in vitro assays

Celsr3$^{R1548Q}$-GFP was generated by QuickChange Mutagenesis from the parent Celsr3-GFP vector (*Chai et al., 2014*). 293 T cells grown on PDL-coated coverslips were transfected with either Celsr3-GFP or Celsr3$^{R1548Q}$-GFP, and analyzed 48 hr later. For analysis of protein localization by immunocytochemistry, cells were briefly fixed in 4% PKS (paraformaldehyde in Krebs + sucrose) for 30 min at room temperature. Cells were then washed three times in PBS for 10 min, blocked for 30 min in PBS + 0.25% TritonX-100, 5% goat serum, then incubated in primary antibody diluted in blocking buffer at 4° overnight. Cells were washed in PBS five times for five minutes each, then incubated in secondary antibody diluted in blocking buffer at room temperature for two hours. Cells were then washed five times for 5 min each in PBS, with DAPI (1:5000) included in the second wash step. Coverslips were then mounted and imaged.

For total and cell surface expression, 293 T cells in 60 mm plates were transfected with either Celsr3-GFP or Celsr3$^{R1548Q}$-GFP and allowed to recover for 48 hr. Cell surface labeling was done with the Pierce Cell Surface Protein Isolation Kit, according to the manufacturer's instructions.

### Quantification and statistical analysis

No statistical methods were used to predetermine sample sizes, but they were similar to our previous work (*Clements et al., 2017*; *Wright et al., 2012*). For all phenotypic analyses, tissue was collected from at least three different offspring obtained from at least three different litters, using at least two different male breeders. All analysis was done blind to genotype. Data was tested for normality and statistical analysis was conducting using JMP Pro version 13.0 (SAS Institute). Comparison between two groups was analyzed using a Student's t test; Comparison between two or more groups was analyzed using a one-way ANOVA and Tukey's post hoc test. *p<0.0001.

## Acknowledgments

We thank members of the Wright laboratory for their assistance and discussion throughout the course of this study; Isabelle Baconguis for assistance with the Laminin-α2-LG4 structure; Marc Freeman, Kelly Monk, Tianyi Mao, Martin Riccomagno and Randal Hand for comments on the manuscript; Krissy Lyons, Jessica Barowski, and Kylee Rosette for technical assistance. We thank David Ginty, in whose lab this work was started, for advice and financial support during the initial phases of the work and for generation of *Celsr3$^{R1548Q}$* mice, and Kevin Campbell for providing *Dag1$^{βcyto}$* mice. This work was supported by NIH grant NS091027 (KMW), the Medical Research Foundation of Oregon (KMW), ARC convention number 17/22–079 (FT), and startup funds from Vollum Institute/OHSU (KMW).

## Additional information

### Competing interests
Fadel Tissir: Reviewing editor, *eLife*. The other authors declare that no competing interests exist.

### Funding

| Funder | Grant reference number | Author |
|---|---|---|
| ARC | 17/22-079 | Fadel Tissir |
| National Institutes of Health | NS091027 | Kevin M Wright |
| Medical Research Foundation | | Kevin M Wright |
| Oregon Health and Science University | | Kevin M Wright |

The funders had no role in study design, data collection and interpretation, or the decision to submit the work for publication.

### Author contributions
L Bailey Lindenmaier, Nicolas Parmentier, Investigation; Caiying Guo, Methodology; Fadel Tissir, Supervision; Kevin M Wright, Conceptualization, Formal analysis, Supervision, Funding acquisition, Investigation, Methodology, Writing—original draft, Project administration, Writing—review and editing

### Author ORCIDs
Kevin M Wright http://orcid.org/0000-0001-5094-5270

### Ethics
Animal experimentation: Mice were handled and bred in accordance with the Oregon Health and Science University IACUC guidelines, protocol #IP00000539.

### Decision letter and Author response
Decision letter https://doi.org/10.7554/eLife.42143.017
Author response https://doi.org/10.7554/eLife.42143.018

## Additional files

### Supplementary files
• Supplementary file 1. Raw data for open book preparation in Dag1 mutants. E12.5 spinal cords were processed for open book preparations and each well-isolated DiI injection site was assessed as showing either normal anterior turning or anterior-posterior randomization.
DOI: https://doi.org/10.7554/eLife.42143.013

• Supplementary file 2. Raw data for open book preparation in Celsr3$^{R1548Q}$ mutants. E12.5 spinal cords were processed for open book preparations and each well-isolated DiI injection site was assessed as showing either normal anterior turning or anterior-posterior randomization.
DOI: https://doi.org/10.7554/eLife.42143.014

• Transparent reporting form
DOI: https://doi.org/10.7554/eLife.42143.015

### Data availability
All data generated or analysed during this study are included in the manuscript and supporting files.

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
