## [Decision Letter]

Thank you for submitting your article "Dystroglycan is a scaffold for extracellular axon guidance decisions" for consideration by *eLife*. Your article has been reviewed by three peer reviewers, and the evaluation has been overseen by a Reviewing Editor and K VijayRaghavan as the Senior Editor. The following individuals involved in review of your submission have agreed to reveal their identity: Chiara Manzini (Reviewer #1); Esther T Stoeckli (Reviewer #2).

The reviewers have discussed the reviews with one another and the Reviewing Editor has drafted this decision, below, to help you prepare a revised submission.

Summary:

We are pleased to write that your manuscript on a role for the transmembrane glycoprotein dystroglycan in axon guidance and tract formation in the mouse spinal cord and forebrain was generally well received. The study nicely follows from your demonstration that dystroglycan is required for commissural axon guidance in the spinal cord (Wright et al., Neuron, 2012) and that it plays a role in retinal ganglion cell axon segregation at the optic chiasm (Clements and Wright, Dev Biol 2018). The reviewers concurred with your conclusions that dystroglycan is required non-cell-autonomously to guide thalamocortical and corticothalamic axons and that dystroglycan appears to act as a scaffold protein without signaling function. A welcome component of the study is that the GPCR adhesion receptor Celsr3 is a new binding partner for dystroglycan controlling commissural axon turning in the spinal cord.

The reviewers and Reviewing Editor consider your study "very interesting and elegant", "a very nice assessment of select guidance events that do and do not involve DG, and of those which do, an in depth analysis of cell autonomous requirements for these DG functions". Results are presented in a clear manner and use multiple genetic (Cre) lines to dissect how dystroglycan acts in different axonal tracts. "It goes beyond distinct aspects of guidance events to give a more global view that will certainly help untangle signaling underlying midline commissural axon guidance (particularly midline crossing and post-crossing signaling or rostral turning), and will also inform future studies on molecular interactions required for correct internal capsule guidance."

Your findings were deemed to be an important contribution to the field because 1) DG is a very complex molecule with high potential for multiple interactions with many distinct guidance systems and extracellular matrix components; 2) they illuminate how cues and receptors are organized and interact outside of the cell; given that signaling though the intracellular domain of dystroglycan is completely dispensable for its role in axon tract formation, dystroglycan is therefore a critical regulator of the extracellular environment; 3) these defects are often overlooked and little studied in congenital dystroglycanopathies.

Essential revisions:

There are a number of aspects in the presentation of the data and its quantification that need amendment.

1) DiI labeling and visualization of axons:a) It is difficult to determine the extent of the various phenotypes at the magnification used; please provide an inset at higher magnification and with better resolution (the *Wnt1^cre^* image in Figure 1F is particularly out of focus).

b) The DiI fluorescence images in Figure 1B, and in Figure 2 and 6, are overexposed and thus not useful for parsing the axonal phenotypes.

2) Expression of dystroglycan in vivo relative to cell bodies and their axons in the spinal cord:

a) Based on Figure 1A, there is no dystroglycan in the cell bodies in the dorsal spinal cord or in the pre-crossing part of the axon. Therefore, the staining of the entire neuron in vitro could be an artifact. Please comment.

b) It is not clear where the dystroglycan regulating axonal pathfinding is coming from. Figure 1 shows dystroglycan in commissural axons in the spinal cord and in the growth cones of cultured neurons, but then you argue with the *Wnt1^cre^* cross that this is completely dispensable.

c) While it may be difficult to pinpoint the specific origin of the dystroglycan without additional genetic crosses in non-neuronal cells, it would be useful to try to demonstrate the expression pattern of dystroglycan at the turning points where these axons stray from their path to support the statement that there is dystroglycan in the neuropil.

d) Likewise, in the Discussion, your suggestion that the basement membrane may be involved; can you show whether and if so, where, the basement membrane is present in relation to the guidance defect?

3) Celsr3 is presented as the binding partner for dystroglycan. Therefore, it would be important to indicate the distribution of Celsr3 expression in the brain with respect to the stages and phenotypes presented in this study.

4) Quantification issues:a) There is a lack of quantification for most all of the phenotypes presented. Please provide N's and some assessment of the penetrance observed in the samples. Unlike the spinal cord results, the aberrant axon guidance phenotypes in the brain are not quantified. In the Materials and methods, you state that at least five embryos from 3 different litters were used, but how many show the aberrant phenotype?

b) The quantification of the spinal cord phenotype is confusing as it is described (Figure 1G, but also in Figure 6D). Where do the error bars come from? When the number of aberrant DiI injection sites is divided by the number of total injection sites, there is no error bar. This type of quantification does not match the description in the Materials and methods section where you state that the quantification was done per spinal cord. One reviewer cannot imagine that it was possible to have more than 50 DiI injection sites per spinal cord!

c) Details on how the quantification have been done for the open books do not match the graph. Please explain and add raw data as a table.

d) Please present the quantification for lack of commissural axon midline stalling defects in the Celsr3 knock-in mutant.

5) Other guidance factors at the spinal cord midline:

Have you looked in a bit more detail at spinal cord commissural axon extension across the midline in the various dystroglycan (and your Celsr3 knockin) mutants? it would be helpful to look at Robo2 and 3 staining in those genetic backgrounds where dystroglycan is thought to not to be required in commissural axons in order to confirm there are indeed no defects in commissural axon midline guidance resulting from cell-autonomous dystroglycan functions and/or dystroglycan cytoplasmic signaling contributions.

6) Celsr3 point mutant (*Celsr3^R1548Q^*) that disrupts its binding to dystroglycan, and that this interaction is specifically required for the proper turning of post-crossing spinal commissural axons.

a) Does AP-Celsr3-LG1 exhibit a binding pattern indicative of association with endogenous DG in the cord and forebrain? Likewise, does AP-Celsr3^R1548Q^ show lack of binding to endogenous DG in sections?

b) in vitro binding studies demonstrate binding between the LG1 but not the LG2 domain of Celsr3 with dystroglycan. Please provide information on possible attempts to use full-length Celsr3 for these interaction studies; if it is not possible to express the full-length protein, it would be good to know this.

---

## [Author Response]

Essential revisions:There are a number of aspects in the presentation of the data and its quantification that need amendment.1) DiI labeling and visualization of axons:a) It is difficult to determine the extent of the various phenotypes at the magnification used; please provide an inset at higher magnification and with better resolution (the Wnt1^cre^ image in Figure 1F is particularly out of focus).b) The DiI fluorescence images in Figure 1B, and in Figure 2 and 6, are overexposed and thus not useful for parsing the axonal phenotypes.

We have provided better, higher resolution images for the DiI labeling experiments in Figures 1, 2, and 6. We have also included high-magnification insets for each of these images to specifically show the axonal guidance defects. In Figures 1 and 6, these insets show the post-crossing trajectory of commissural axons in both the anterior and posterior directions. In Figures 2 and 6, the insets show the target areas of the labeled axonal populations (lateral cortex for experiments in which ascending thalamocortical axons are labeled, or the thalamus in experiments where descending corticothalamic axons are labeled). While the axons in Figure 1D still appear a little out of focus, this is because we cannot remove the meninges in *DG^Flox/-^; Sox2^Cre^* mutants due to their lack of structural integrity. Regardless, the anterior-posterior randomization of axons can clearly be seen in Figure 1D.

2) Expression of dystroglycan in vivo relative to cell bodies and their axons in the spinal cord:a) Based on Figure 1A, there is no dystroglycan in the cell bodies in the dorsal spinal cord or in the pre-crossing part of the axon. Therefore, the staining of the entire neuron in vitro could be an artifact. Please comment.

We have provided an additional image of Dystroglycan staining in the spinal cord (Figure 1—figure supplement 1A), in which staining can also be seen in pre-crossing commissural axons. We are confident of the specificity of this staining, because it is completely gone in mice lacking the cytoplasmic domain of Dystroglycan (Figure 1—figure supplement 1B). While we agree that the staining is much higher in the cell body of the cultured commissural neurons in Figure 1B than might be expected based on intact tissue sections, the main point of showing the cultured neurons is to support the observation that Dystroglycan is expressed in the axons of spinal commissural neurons. If necessary, we can remove the in vitro data (Figure 1B).

b) It is not clear where the dystroglycan regulating axonal pathfinding is coming from. Figure 1 shows dystroglycan in commissural axons in the spinal cord and in the growth cones of cultured neurons, but then you argue with the Wnt1^cre^ cross that this is completely dispensable.

We apologize for the lack of clarity here. Indeed, our analysis indicates that Dystroglycan is present in both the commissural axons (Figure 1A, B), as well as the surrounding environment that these axons grow in, particularly along the basement membrane (Figure 1A, inset). We were unsure of where the Dystroglycan regulating the axonal pathfinding was coming from (i.e.: cell autonomously in axons or non-cell autonomously from the surrounding environment and basement membrane). That is why we turned to genetic studies. Our interpretation of the normal commissural crossing phenotypes in *DG; Wnt1^Cre^*mutants is that despite being expressed in commissural axons, Dystroglycan is not required within these axons themselves, and therefore is functioning non-cell autonomously in this context. We have made text changes to the Results section to clarify this point (Subsection “Dystroglycan functions non-cell autonomously as an extracellular scaffold 104 to guide commissural axons”, first paragraph).

c) While it may be difficult to pinpoint the specific origin of the dystroglycan without additional genetic crosses in non-neuronal cells, it would be useful to try to demonstrate the expression pattern of dystroglycan at the turning points where these axons stray from their path to support the statement that there is dystroglycan in the neuropil.

In the new images in Figure 1—figure supplement 1A and for Figure 1A we provide a higher-magnification inset showing Dystroglycan expression along the basement membrane immediately adjacent to the floor-plate where commissural axons initiate their anterior turn.

d) Likewise, in the Discussion, your suggestion that the basement membrane may be involved; can you show whether and if so, where, the basement membrane is present in relation to the guidance defect?

In our previous paper (Wright, et al., Neuron 2012; Supplementary Figures 5A and 6E), we showed that commissural axons grow along laminin concentrated in the spinal cord basement membrane (Supplementary Figure 6E) and that in mice lacking functional Dystroglycan, the basement membrane is fragmented in the areas where axon guidance defects are observed (Supplementary Figure 5A). We had referenced this at the beginning of the Results section, but neglected to reference it again at this point in the Discussion, and have corrected that oversight.

3) Celsr3 is presented as the binding partner for dystroglycan. Therefore, it would be important to indicate the distribution of Celsr3 expression in the brain with respect to the stages and phenotypes presented in this study.

Previous work has nicely detailed the expression pattern of Celsr3 mRNA in both the brain and spinal cord at the specific areas and stages of development we analyzed in our manuscript. We neglected to reference this data in the initial submission, and have corrected it (subsection “Dystroglycan binds to the axon guidance receptor Celsr3”).

4) Quantification issues:a) There is a lack of quantification for most all of the phenotypes presented. Please provide N's and some assessment of the penetrance observed in the samples. Unlike the spinal cord results, the aberrant axon guidance phenotypes in the brain are not quantified. In the Materials and methods, you state that at least five embryos from 3 different litters were used, but how many show the aberrant phenotype?

We have provided N’s in the figure legends for all phenotypic analysis, and have clarified that the brain phenotypes are fully penetrant for the Dystroglycan mutants in the text of the Results section (subsection “Dystroglycan is required for axon tract development in the forebrain”, first paragraph).

b) The quantification of the spinal cord phenotype is confusing as it is described (Figure 1G, but also in Figure 6D). Where do the error bars come from? When the number of aberrant DiI injection sites is divided by the number of total injection sites, there is no error bar. This type of quantification does not match the description in the Materials and methods section where you state that the quantification was done per spinal cord. One reviewer cannot imagine that it was possible to have more than 50 DiI injection sites per spinal cord!

We apologize for the confusion, and have clarified how the open book preparations were quantified in the Materials and methods section, Results and figure legends. This is done the standard way of quantifying open book preparations to our understanding. Briefly, for each animal, well isolated DiI injection sites (ranging from 3-13 per animal) were scored for normal anterior turning or AP randomization. These results were then used to calculate the percentage of sites showing correct anterior turning for that animal. Therefore, each animal represents a single “N”. The N’s for each genotype are then averaged, and SEM is calculated.

c) Details on how the quantification have been done for the open books do not match the graph. Please explain and add raw data as a table.

We have now included tables with the raw data for open book experiments.

d) Please present the quantification for lack of commissural axon midline stalling defects in the Celsr3 knock-in mutant.

We did not observe any stalling in the *Ceslr3^R1548Q^* mutants, so there is nothing to quantify. This has been clarified in the text of the Results section.

5) Other guidance factors at the spinal cord midline:Have you looked in a bit more detail at spinal cord commissural axon extension across the midline in the various dystroglycan (and your Celsr3 knockin) mutants? it would be helpful to look at Robo2 and 3 staining in those genetic backgrounds where dystroglycan is thought to not to be required in commissural axons in order to confirm there are indeed no defects in commissural axon midline guidance resulting from cell-autonomous dystroglycan functions and/or dystroglycan cytoplasmic signaling contributions.

The reviewers raise a good point that our initial analysis only used DiI labeling in open book preparations to examine commissural phenotypes in the mutant mice.We have examined spinal cord sections in the various mutants with antibodies to Robo1 and Robo2, as well as L1 to label post-crossing commissural axons. This data confirms the lack of any commissural axon guidance defects in *DG^Flox/-^; Wnt1^Cre^* and *DG^βcyto/-^*mutants, supporting the findings obtained with DiI open-book preparations. We have included this data in Figures 1—figure supplement 1C-F and Figure 6—figure supplement 1D-E.

6) Celsr3 point mutant (Celsr3^R1548Q^) that disrupts its binding to dystroglycan, and that this interaction is specifically required for the proper turning of post-crossing spinal commissural axons.a) Does AP-Celsr3-LG1 exhibit a binding pattern indicative of association with endogenous DG in the cord and forebrain? Likewise, does AP-Celsr3^R1548Q^ show lack of binding to endogenous DG in sections?

This is an excellent suggestion, and we have now done this experiment and include the results in Figure 5E-J. We find that AP-Ceslr3-LG1 binding is enriched in post-crossing commissural axons in E12 spinal cord and axons in the internal capsule at E14, consistent with the expression of Dystroglycan in these regions (Figures 1A, Figure 1—figure supplement 1A, and Figure 2—figure supplement 1A-B). Furthermore, AP-Ceslr3-R1456Q shows reduced binding in these regions, consistent with our in vitro binding results using purified proteins (Figure 5D). This data strengthens the observation that Celsr3 binds to Dystroglycan in vivo, and that the R1548Q mutation disrupts this binding.

b) in vitro binding studies demonstrate binding between the LG1 but not the LG2 domain of Celsr3 with dystroglycan. Please provide information on possible attempts to use full-length Celsr3 for these interaction studies; if it is not possible to express the full-length protein, it would be good to know this.

Celsr3 is a very large protein (~360kD) that proved to be rather difficult to work with. We did attempt to generate full-length ectodomain of Ceslr3 (~280kD), as well as versions that contained both the LG1 and LG2 domains and their surrounding regions or versions that lacked just the cadherin repeats. However, these constructs appeared to be generally unstable and we could not generate sufficient quantitates of secreted AP-tagged protein for binding studies. We have now indicated this in the text of the manuscript (subsection “Dystroglycan binds to the axon guidance receptor Celsr3”, third paragraph).